# On the selection of neural architectures from a supernet

Gabriel Meyer-Lee[1]  Nick Cheney[1]

[1]University of Vermont, Computer Science Department

**Abstract**  After DARTS provided a method utilizing a supernet to search for neural network architectures entirely through gradient descent, differentiable supernet-based methods emerged as a powerful and popular approach to efficient neural architecture search (NAS). Following works improved upon many aspects of the DARTS algorithm but generally kept the original method of selecting the final architecture, pruning the lowest magnitude architecture weights, though critiques of this approach have led to alternative architecture selection mechanisms, such as a perturbation-based method. Here we perform a broad comparative evaluation of architecture selection methods in combination with different techniques for training the supernet, and highlight the interdependence between various methods for supernet training and architecture selection mechanisms. We show the potential for improved results for many NAS supernet training methods via alternate architecture selection mechanisms relative to the pruning-based architecture selection they were introduced, and are typically evaluated, with. In evaluating architecture selection methods, we also demonstrate how zero-shot NAS methods may be effectively integrated into supernet NAS training as new architecture selection mechanisms.

## 1 Introduction

Neural architecture search (NAS) names the problem of automatically selecting an effective neural network architecture for given data. NAS is a subset of automated machine learning (AutoML) that is focused on the topology and operations of deep learning models. In their survey of work on NAS, Elsken et al. (2019) establish a general model of three components constituting the NAS process: a search space to define the set of possible architectures, a search strategy to traverse over potential solutions, and a performance estimation strategy to rate the quality of solutions considered by the search strategy.

Early NAS algorithms required thousands of GPU hours to learn weights for each candidate network in order to obtain an estimate of that candidate's performance. Researchers soon developed techniques to obtain performance estimates more rapidly, including weight inheritance (Real et al., 2017), predictive models (Liu et al., 2018), shortened training (Zoph et al., 2018), and weight-sharing (Pham et al., 2018). The weight sharing technique, originally part of ENAS, allowed performance estimates for any candidate network to be obtained after training a single neural network. This technique was based on training a supernet, which represented a superposition of all candidate networks. DARTS then demonstrated a gradient-based method to learn architecture weights attached to the supernet, which collapsed the process of performance estimation and search into the training of a single neural network (Liu et al., 2019). Thus, these new efficient NAS methods no longer fit neatly into the existing general theoretical framework, making it difficult to interpret how new innovations affect the performance estimation and searching capabilities of the algorithm.

A more recent survey by Ren et al. (2021) of NAS discusses the range of methods that have been developed from the concepts in DARTS. In this work, we perform our own analysis of the supernet methods reviewed in Ren et al. (2021) and two diverging lines of research, represented by the DARTS-like and "true one-shot" approaches to supernet NAS. Our review reveals the DARTS-like algorithms to be more popular in the literature, made clear by the widespread use of the architecture selection method utilized by DARTS, magnitude-based pruning of the architecture weights.

This architecture selection method has been previously criticized as ineffective for the algorithm (Li and Talwalkar, 2019; Yu et al., 2020; Yang et al., 2020). Some studies suggest the architecture weight learning process overvalues specific operations, such as skip connections (Wang et al., 2021), while others attribute the issue to the fact that DARTS does not evaluate actual candidate architectures during training, only continuous combinations of them (Xie et al., 2019). Most of these works propose techniques to train the architecture weights in order to overcome these issues. But, DARTS-PT (Wang et al., 2021) instead rejects pruning in favor of a perturbation-based architecture selection method and goes so far as to question the utility of the rest of the DARTS algorithm. We perform a case study to examine these findings, showing that minimal variation in how the supernet is trained will affect the trends observed and the conclusions drawn by the original study. We then expand this approach to a holistic evaluation of architecture selection in supernet NAS.

A recent line of NAS research, the "zero-shot" methods, obtain even faster runtimes than the supernet methods by not training even a single model. Rather, they obtain performance estimates based on metrics computed from the structure of candidate networks (Abdelfattah et al., 2021). Typically these metrics are used on an untrained supernet but we provide a new reframing as a method of selecting architectures from a trained supernet and evidence in their effectiveness.

Here, we contribute a holistic and systematic analysis that evaluates combinations of supernet training and architecture selection techniques across a range of training contexts. This allows us to address the question of when and why magnitude-based pruning fails, as well as to analyze how different components of the training context affect the evaluation of NAS algorithms. For example, both operation bias and the discretization loss (White et al., 2023; Xie et al., 2020) have been suggested to factor in the the failure modes of magnitude pruning. On NAS-Bench-201, operation bias has a dominant negative impact on the results of DARTS, however, the DARTS- (Chu et al., 2021a) results reveal that correcting for this alone without addressing the discretization error will still leave magnitude pruning suboptimal. In contrast, the success of DirichletNAS is explained by it's small discretization gap and lack of bias toward skip connections.

We do not demonstrate any single alternative architecture selection technique to be clearly superior to magnitude-based pruning but instead show that perturbation, sampling, and zero-shot methods can all be highly effective in different training contexts. While being able to point out a single optimal approach would be helpful for advancing NAS, we believe that that simply highlighting this complexity is important given the current tendency of papers introducing novel supernet training methods to benchmark their approach using just a single architecture selection methods (and often magnitude-based pruning; Appendix Figure 5). While we don't believe it necessary for every use-case of supernet NAS to take on the expense of evaluating every supernet training method with every architecture selection mechanism, our findings here suggest that increasing the use of this approach when benchmarking new, or existing, algorithms would better contextualize (and perhaps more fairly evaluate) different approaches – while also being able to improve the stated performance metrics for new proposed supernet training algorithms.

The primary contributions of this work are thus to demonstrate and highlight that:

- Supernet NAS methods are often published with architecture selection methods that are in many contexts sub-optimal (Sec. 5.1)

- Zero-shot NAS approaches can be highly successful for architecture selection using trained supernets, though this is not how they are typically used (Sec. 4.1 and 5)

- Operation bias and discretization loss together can explain the failure modes of supernet NAS methods but either alone can be misleading (Sec. 5.3)

- A systematic comparison of combinations suggests that there are not necessarily clear takeaways and easy predictions for which arch selection mechanism will be best, but the best supernet training algorithms are successful with multiple selection methods (Sec. 5)

## 2 Related Work

### 2.1 Differentiable supernet NAS

DARTS was among the first differentiable NAS algorithm, utilizing a "continuous relaxation" method to parametrize the set of possible architectures, by placing trainable weights on each possible operation (Liu et al., 2019). The final model is selected from these trained architecture weights by simply keeping only the operations with the largest weight ("magnitude-based pruning").

In motivating SNAS, Xie et al. (2019) comment on the performance estimation aspect of DARTS, demonstrating the disparity in the validation accuracy obtained from the full shared-weight model and the pruned final architecture. SNAS addresses this limitation by sampling individual ("softened") architectures, while GDAS samples discrete architectures (Dong and Yang, 2019c) without softening. DrNAS substitutes a dirichlet distribution over the architecture weights (Chen et al., 2021b). These stochastic methods preserve the gradient-based approach of DARTS, while proposing a specific (and increasingly discrete) architecture at each step.

While all of the algorithms described above would be considered DARTS-like methods, Bender et al. (2018) demonstrated an alternative approach, implementing a shared-weight supernet trained via path dropout and demonstrating a strong correlation between performance estimates from shortened training and the shared-weight model. The "true one-shot" methods follow this approach, training a supernet without even the implicit search over candidate architectures obtained via architecture weights (for example, by sampling subnetworks uniformly to train).

### 2.2 Zero-shot NAS

Recently, methods have been proposed to select an architecture without training even a single architecture model. Much of these methods are based on proxies, like EcoNAS (Zhou et al., 2020), while TE-NAS (Chen et al., 2021a) demonstrated a zero-shot search based on measures emerging from deep learning theory. Abdelfattah et al. (2021) suggest as potentially useful in NAS a range of measures that have been used to compute saliency in other deep learning problems.

In our project we propose a new unification of zero- and one-shot NAS methods, which we demonstrate by modifying the adapted synflow (Tanaka et al., 2020) measure utilized by Abdelfattah et al. (2021) as well as their implementation of the Jacobian covariance measure developed by Mellor et al. (2021) to function as architecture selection techniques on a supernet. The initialized architectures that these metrics are computed on can be viewed as subnetworks drawn from an untrained supernet. We generalize this to include trained supernets as well.

### 2.3 Supernet NAS meta-analysis

The literature review of Ren et al. (2021) expands on, and reframes, the general model of NAS proposed by Elsken et al. (2019), adding the selection and evaluation of the final architecture to the theoretical model. This parallels the conceptual model advanced in our project, however our framework is adjusted to suit the specific form of supernet NAS algorithms.

Li and Talwalkar (2019), in a key early paper demonstrating meta-analysis of NAS algorithms, show that the selection of search space can have a greater effect on the final performance than the selection of NAS algorithm. Our paper continues in this line of meta-analysis, exemplified by our contextualization of algorithm performance in Sec. 5.

Yu et al. (2020) initiated the critical inquiry into the use of weight-sharing for performance estimation by evaluating popular weight-sharing NAS algorithms in a reduced search space of 32 architectures. They demonstrated a lack of correlation in ranking between the performance estimates obtained from the shared-weight model and the train-from-scratch test accuracy. Further research evaluating the rankings of small samples (Yang et al., 2020) or small search spaces (Zhang et al., 2020) showed high variance in rankings across random seeds after training the shared weight model through random sampling. Like these prior studies, we also suggest the accuracy of the top

| Search Space | DARTS | | | DARTS- | | |
|---|---|---|---|---|---|---|
| | Max $\alpha$ | PT | PT w/ fix $\alpha$ | Max $\alpha$ | PT | PT w/ fix $\alpha$ |
| NB-201 | 45.7 | 11.89 | **6.20** | $6.75 \pm 0.35$ | **$6.56 \pm 0.33$** | $7.22 \pm 0.95$ |
| S1 | 3.84 | 3.50 | **2.86** | $2.84 \pm 0.26$ | **$\underline{2.83 \pm 0.15}$** | $2.89 \pm 0.15$ |
| S2 | 4.85 | 2.79 | **2.59** | $2.74 \pm 0.15$ | **$\underline{2.46 \pm 0.08}$** | $2.73 \pm 0.13$ |
| S3 | 3.34 | $\underline{2.49}$ | 2.52 | $2.62 \pm 0.08$ | $2.62 \pm 0.10$ | **$2.60 \pm 0.13$** |
| S4 | 7.2 | 2.64 | **2.58** | $3.54 \pm 0.33$ | $2.63 \pm 0.04$ | **$\underline{2.53 \pm 0.09}$** |

Table 1: A case study showing the importance of separately evaluating supernet training and architecture selection. The DARTS-PT architecture selection method run on a supernet trained on standard DARTS *(left)* tends to perform best when not training architecture weights ("Perturb w/ fix $\alpha$" is best in 4 of 5 search spaces). But when the same perturbation method is applied to a supernet trained via DARTS- *(right)*, performance improves when including architecture weights (as "Perturb w/ fix $\alpha$" is best in just 1 of 5 search spaces). This reversal demonstrates that the perturbation method of selecting architectures is reliant on the particular training of the supernet. The top architecture selection technique for the version of DARTS is indicated in bold while the overall top method for each search space is underlined. Search spaces are explained in Appendix C.

model alone can be a misleading evaluation technique for NAS algorithms. However, our focus is not on design issues with specific components of the NAS process, but on the (perhaps unintended) interactions between the proposed innovation and the other NAS components. The closest prior work to our study is that of Ning et al. (2021), as they also use comparative evaluation experiments to study the behavior of supernet NAS algorithms. In this study, however, we focus on architecture selection rather than performance prediction.

In this work, we build on these prior works of meta-analysis of NAS to clarify the conceptual structure of supernet NAS algorithms and use this framework to understand their performance. We use a literature review and an experimental case study to highlight the need for architecture selection and supernet training to be studied as interdependent processes. We perform independent and broad comparative evaluations of these algorithms, in order to shed light on competing theories of issues with architecture selection and performance estimation of supernet NAS. In the process of this evaluation, we synthesize a novel NAS algorithm (combining DARTS-like training with evaluations from zero-shot methods), not to propose the algorithm but as a proof of concept of the design space supported through our general model of supernet NAS.

## 3 Case Study: two modifications to DARTS

Two recent papers demonstrated non-overlapping modifications to DARTS named DARTS- (Chu et al., 2021a) and DARTS-PT (Wang et al., 2021). Each paper aimed to overcome the problem of unstable architecture weights in DARTS. DARTS- addressed this issue by adding an auxiliary skip connection to each mixed operation in DARTS, aiming to stabilize the training of architecture weights. DARTS-PT instead dismissed the architecture weights entirely, using a perturbation based method of selecting operations after training the supernet. The authors conclude that the perturbation technique may work better if DARTS is trained without architecture weights at all as they were able to demonstrate a smaller error using their method on a supernet trained with fixed architecture weights in comparison with one trained by DARTS. This result would suggest that the core innovation of DARTS—the continuous relaxation technique for approximating gradient updates to the architecture of the model—may not be of significant use[1].

---

[1]A surprising and provocative implication, highlighted as an ICLR 2021 Outstanding Paper Award

By implementing DARTS- within the provided codebase for DARTS-PT and replicating their tests on CIFAR10, we observe a near reversal of this result (Table 1). Once training of the architecture weights is stabilized via the auxiliary skip connection, perturbation-based selection with a supernet trained without architecture weights is no longer the dominant method.

The perturbation method still appears quite effective overall, although it now benefits from training the architecture weights ($\alpha$). Interestingly, the perturbation method does not utilize the architecture weights for architecture selection itself, so this result indicates that properly trained architecture weights influence the training of the shared weights in a way that can assist the perturbation process. This hints at the dependence of shared weights on architecture weights, while the overall reversal of the effectiveness of DARTS-PT suggests a dependence of an architecture selection method on the training of the supernet model to which it is applied.

DARTS-PT demonstrated an innovative solution to the instability in DARTS by intervening at the time of architecture selection rather than during training. In Sections 4.1 and 5 below, we extend this approach to study architecture selection more broadly. The above case study demonstrates both that some architecture selection algorithms (e.g. pruning) can make up for limitations of specific supernet training algorithms (e.g. instability in DARTS), and also that this relationship between supernet training and architecture selection means that using a single approach to either may not reliably assess the other's potential effectiveness. For these reasons, we evaluate a variety of supernet training and architecture selection algorithms in isolation and combination to understand the effectiveness of each algorithm and the interactions between them.

## 4 Architecture selection in supernets

Prior work outlines several explanations for the failure of magnitude-based pruning in supernets, we highlight two distinct explanations particularly tied to architecture selection: operation bias and discretization loss (White et al., 2023). The operation bias describes the tendency of the supernet training process to assign disproportionate weight to a specific operation, in this case the skip-connection, compared the the actual utility of the operation. This is explained as byproduct of the greater utility of the skip connection in avoiding vanishing gradients during supernet training compared to the utility in training the smaller candidate architectures (Chu et al., 2021a; Wang et al., 2021). The discretization loss describes the drop in accuracy observed when using supernet shared-weights optimized for continuous architectural weights with discrete architectures. This has been shown to be a source of instability in DARTS, where the disconnect between the supernet training process and the final "hard prune" means that the performance of the supernet has little relation to the that of the final architecture selected and the interdependence between candidate operations is ignored during selection (Xie et al., 2019; Li and Talwalkar, 2019).

The supernet training methods we select for experimentation align with distinct modifications to DARTS designed to address these sources of failure. DARTS- is a variation which directly attempts to address operation bias. SNAS, GDAS, and DrNAS all attempt to mitigate discretization loss through the use of sampling during training, with GDAS actually sampling discrete architectures and SNAS and DrNAS sampling softened architectures, which approach discrete samples over the course of supernet training by decaying a temperature parameter. The stochasticity of these algorithms may also provide some regularization capable of decreasing operation bias. While prior studies have shown both of these phenomena to be significant issues for supernet NAS methods, we study how they interact.

In order to do so, we focus on the process of architecture selection, emphasizing that it is not merely a trivial discretization step at the end of supernet training, but an architecture search itself which takes in the supernet model and associated weighted distribution over the architecture space and selects the single top architecture for that search space. We utilize the original names of the algorithms to denote only methods for shared-weight model training (e.g. DARTS, SNAS), while

we separate out architecture selection methods and name them based on the criterion used (e.g. prune, perturb).

We review the architecture selection methods demonstrated in literature in detail in Appendix D. From these, in addition to the commonly used magnitude-based **pruning**, we test the basic method utilized for one-shot supernet NAS of sampling a set of candidate architectures (in our case, using the architecture weights) and ranking the candidates by validation accuracy as estimated using the supernet's shared-weights, referring to this method as **valid** (Bender et al., 2018). Taking an open-ended view of architecture selection, we also implement methods used to rank sample architectures not generally applied to trained supernets, the zero-shot NAS metrics of **synflow** and Jacobian covariance (**jacob** or **jacob cov**) (Tanaka et al., 2020; Mellor et al., 2021). Lastly, we test the perturbation-based technique developed by Wang et al. (2021) specifically to overcome the issue of operation bias in DARTS (**perturb**).

### 4.1 Evaluating architecture selection

As we have framed architecture selection as search algorithms that take a supernet as input and output a single selected architecture, we can compare these algorithms in isolation from the process of supernet training via their final architectures if we can produce a supernet trained in a sufficiently neutral manner to suffice as a baseline.

We consider two training (or lack thereof) approaches for developing synthetic supernets: one randomly initialized and untrained ("untrained") and one trained via sampled discrete architectures as with Li and Talwalkar (2019), except with biased oversampling of the top architectures (inversely proportional to their train-from-scratch test performance) rather than their uniform sampling of architectures ("informed"). Then, we consider two approaches to obtaining samples of candidate architectures, using a uniform distribution over the architecture space ("uniform") and using the same biased sampler used to train the supernet detailed above ("informed"). We combine these to create a total of four synthetic supernets, as described in further detail in Appendix E.

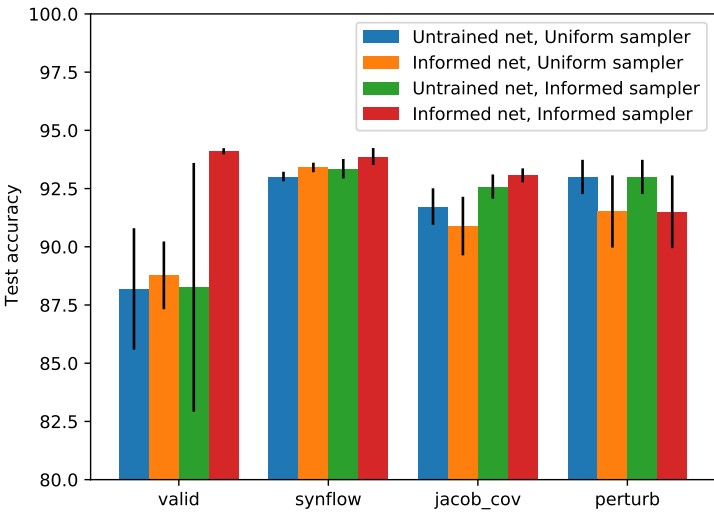

Figure 1: Comparison of architecture selection performance on CIFAR-10 with given baseline models. Baseline models are constructed by combining two components: a supernet, with shared weights which are either randomly initialized and *untrained* or have been training by sampling individual candidate networks using an *informed* sampler, and an architecture sampler, which uses either a *uniform* distribution or the *informed* sampler.

Comparing the results using the untrained vs trained supernets both with samples drawn from the informed sampler, we note that both methods originally proposed for use in NAS as zero-shot algorithms seem to benefit slightly from having effectively trained weights – suggesting their potential use as architecture selection mechanisms – despite their use-cases in the past being solely for zero-shot NAS. Synaptic flow improves in test accuracy from 93.35 ± 0.42 to 93.88 ± 0.36, and Jacobian covariance improves from 92.50 ± 0.52 to 93.05 ± 0.43 on CIFAR-10.

For the perturbation method of architecture selection, training the shared-weight model actually decreases performance, dropping from 93.00 ± 0.73 with randomly initialed weights to 90.50 ± 0.22 with trained shared-weights

This mirrors the results of the case study shown in Table 1, but in this case we present a more extreme (though similarly narrow) result: the perturbation method isn't just doing better without architecture weights than with them, but, in comparison to our baseline method of shared weight training, perturbation-based architecture selection shows an improvement from not training the supernet at all.

The results for selecting architectures based on validation accuracy are clear: this method is only effective with the trained supernet and the informed sampler. It is fairly straightforward that using the same distribution of architectures used to train the supernet to draw a sample will lead to the most reliable validation accuracies across the sample. Without a trained supernet, validation accuracy measurement is largely uninformative and when the architectures validated are drawn from a different distribution, so they do not match the architectures the weights are trained for, the resulting validation accuracy measurements may be quite noisy.

These trends are fairly consistent when tested on CIFAR-100 as well, shown in Appendix E.

## 5 Combined Evaluation

As our case study suggested, the conditions for failure of architecture selection from a supernet are highly dependent on the specifics of the training process of the supernet. By exhaustively analyzing each of the selected supernet training algorithms in combination with each architecture selection technique, we can observe both the failure modes of individual supernet training algorithms and broad trends in the performance of supernet NAS algorithms based on their design. Table 2 shows these combined NAS algorithms for various lengths of training, datasets, and search spaces, reporting average train-from-scratch test accuracy across three independent trials. The top performing supernet training algorithms for a variety of architecture selection methods were SNAS and Dirichlet NAS. They made up the the winning combination (bold cells in table) in 10 of the 11 search space/dataset/training length contexts. The architecture selection method that best paired with them was less consistent, although pruning was the most effective in 5 of those 10 contexts.

### 5.1 When should we not prune?

#### 5.1.1 DARTS and DARTS- on NAS-Bench-201.
DARTS with pruning is notably ineffective on NAS-Bench-201. DARTS- with pruning performs significantly better in this search space across the board, indicating that a substantial component of DARTS test error can be attributed to DARTS overvaluing skip connections–as suggested in Chu et al. (2021a). In all but one training context of NAS-Bench-201, pruning on DARTS is outperformed by every other architecture selection algorithm studied. This is not the case for DARTS-. However, pruning on DARTS- is still outperformed in every NAS-Bench-201 training context by at least one architecture selection algorithm. This shows that adding the extra skip connection does not fully correct DARTS' failure on NAS-Bench-201.

#### 5.1.2 GDAS with shorter training durations.
On NAS-Bench-201, pruning is the most effective architecture selection algorithm for GDAS only when the supernet is trained for 250 epochs. On CIFAR10 and CIFAR100, when the supernet is trained for less than 250 epochs, almost all other architecture selection algorithms are superior to pruning. One possible hypothesis for this is that the discrete

| Search space | NASBench-201 | | | | | | | DARTS-space | | | |
|---|---|---|---|---|---|---|---|---|---|---|---|
| Data set | CIFAR10 | | | CIFAR100 | | | IN16 | CIFAR10 | | CIFAR100 | |
| Epochs | 50 | 100 | 250 | 50 | 100 | 250 | 100 | 50 | 100 | 50 | 100 |
| darts/prune | 45.70 | 40.16 | 15.80 | 61.03 | 61.03 | 45.36 | 81.59 | 3.27 | 3.08 | 16.67 | 17.03 |
| darts/valid | 14.67 | 9.07 | 11.48 | 50.98 | 36.41 | 47.47 | 79.21 | 3.48 | 3.16 | 17.88 | 18.60 |
| darts/synflow | 6.79 | 9.48 | 9.35 | 30.12 | 33.24 | 42.80 | 57.41 | 3.66 | 3.52 | 17.64 | 19.43 |
| darts/jacob | 11.38 | 10.03 | 9.55 | 33.15 | 36.44 | 35.91 | 71.52 | 3.01 | 3.04 | 17.60 | 17.56 |
| darts/perturb | 11.77 | 14.34 | 14.31 | 40.43 | 44.13 | 42.92 | 65.04 | 3.25 | 3.01 | 17.50 | 16.87 |
| darts-/prune | 6.31 | 9.36 | 8.30 | 32.76 | 32.46 | 32.08 | 77.03 | 2.66 | 2.89 | 17.02 | 19.89 |
| darts-/valid | 7.48 | 11.90 | 8.75 | 38.15 | 44.68 | 33.95 | 77.49 | 2.98 | 2.80 | 17.09 | 19.04 |
| darts-/synflow | 6.81 | 6.29 | 6.31 | 29.15 | 28.51 | 31.18 | 54.68 | 3.22 | 3.03 | 17.48 | 19.81 |
| darts-/jacob | 8.01 | 8.16 | 6.00 | 28.99 | 32.18 | 31.12 | 61.61 | 3.11 | 3.08 | 16.98 | 19.08 |
| darts-/perturb | 6.23 | 10.17 | 8.12 | 28.10 | 35.11 | 30.18 | 64.65 | 2.72 | 2.93 | 16.92 | 17.36 |
| snas/prune | 5.71 | 5.92 | 6.10 | **27.23** | 28.37 | 30.40 | 53.66 | 3.10 | 3.16 | **16.24** | 18.22 |
| snas/valid | 14.84 | 8.31 | 7.51 | 41.31 | 36.95 | 36.49 | **53.44** | 3.19 | 3.21 | 16.79 | 18.48 |
| snas/synflow | 6.57 | 6.36 | 6.22 | 29.71 | 29.08 | **28.43** | 56.93 | 3.61 | 3.23 | 18.32 | 18.19 |
| snas/jacob | 7.87 | 7.71 | 7.68 | 30.11 | 32.47 | 32.69 | 54.64 | 3.13 | 3.20 | 16.41 | 17.29 |
| snas/perturb | 6.19 | 7.24 | 7.28 | 28.01 | 33.65 | 32.36 | 56.20 | 2.96 | 2.96 | 17.74 | 16.96 |
| gdas/prune | 34.76 | 12.55 | 6.60 | 73.33 | 83.58 | 29.54 | 58.98 | 3.04 | 2.66 | 16.49 | **16.69** |
| gdas/valid | 21.48 | 17.40 | 9.40 | 57.00 | 52.97 | 43.80 | 69.26 | 4.01 | 3.55 | 17.68 | 18.49 |
| gdas/synflow | 7.03 | 7.03 | 6.66 | 30.82 | 30.21 | 30.45 | 58.62 | 4.94 | 4.24 | 20.36 | 20.31 |
| gdas/jacob | 7.50 | 9.30 | 8.54 | 32.69 | 36.20 | 34.95 | 63.39 | 3.17 | 3.62 | 17.65 | 17.77 |
| gdas/perturb | 11.09 | 8.25 | 7.67 | 38.41 | 38.81 | 31.35 | 61.41 | 2.90 | 3.30 | 17.48 | 17.21 |
| drnas/prune | 5.84 | **5.64** | 17.54 | 28.03 | 28.62 | 45.36 | 53.66 | **2.65** | **2.60** | 16.73 | 17.96 |
| drnas/valid | 6.32 | 6.35 | 6.40 | 32.42 | 30.03 | 53.44 | 54.68 | 3.36 | 2.74 | 17.25 | 17.55 |
| drnas/synflow | 6.40 | 6.04 | **5.80** | 28.85 | **27.48** | 29.33 | 53.66 | 3.67 | 3.46 | 18.26 | 17.06 |
| drnas/jacob | 8.31 | 7.35 | 10.42 | 32.88 | 34.44 | 34.71 | 56.15 | 3.21 | 2.84 | 17.26 | 17.12 |
| drnas/perturb | **5.66** | 5.88 | 11.14 | 28.42 | 29.29 | 33.30 | 56.04 | 2.78 | 2.77 | 17.08 | 17.10 |

Table 2: Test error values for all supernet training and architecture selection algorithm combinations, evaluated across different search spaces, datasets, and training lengths. The bolded values represent the lowest error attained in each training context and the underlined values indicate other combinations that outperformed pruning for a given supernet training algorithm in that training context.

architectures sampled by GDAS provide sparser gradients than the updates on softened samples or full shared-weight models from other algorithms. This results in the need for additional training time to produce sufficiently informative architecture weight magnitudes.

## 5.2 DARTS-space

While many supernet training algorithms demonstrated alternative architecture selection methods surpassing pruning in different training contexts within the DARTS-space, the very highest accuracy architecture discovered in each training context was produced by magnitude-based pruning. One key explanation for this is the decreased impact of operation bias in the larger, more complex DARTS search space. This is empirically evident from the results using CIFAR 10, where the top architectures discovered by DrNAS actually had more skip connections than the slightly lower performing methods discovered by other supernet training algorithms.

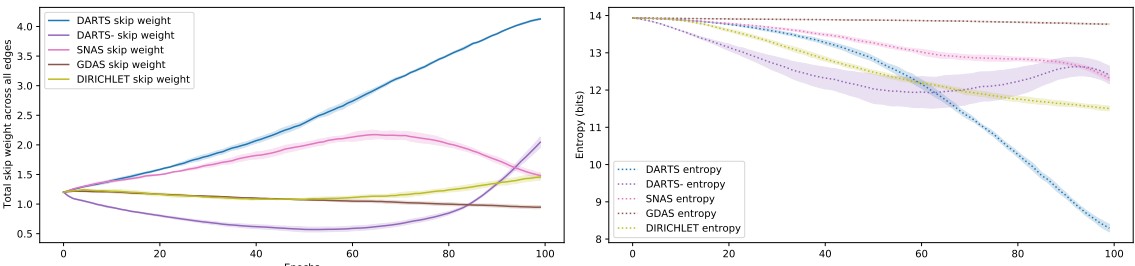

Figure 2: *Left:* The total weight (after softmax normalization) assigned to skip connections across all edges by each supernet training algorithm during training. *Right:* The entropy (summed over all edges) of the architecture weights for eah supernet training algorithm during training. Both plots show results for training with CIFAR 10 in the NAS-Bench-201 search space.

## 5.3 Explaining performance

DARTS dramatic failure on NAS-Bench-201 can largely be explained as a result of operation bias. Viewing the weight assigned to the skip connection operations by each supernet training algorithm reveals that, in this search space, DARTS collapses on a model a largely composed of skip connections, as exemplified in Figure 2. In one sense, this results in a small discretization gap. As indicated by the entropy plots, due to this major overweighting of skip connections the difference between the architecture weights and final selected architecture is significantly smaller for DARTS than the other supernet training algorithms. However, this small discretization gap actually results in a huge discretization loss as nearly all of the weighted operations which are actually transforming the cell's input are removed. The top performing supernet training method in this context, DrNAS, for which pruning is the best architecture selection technique in the context, demonstrates the second lowest entropy after DARTS without very high weight assigned to skip connections, indicating success as the result of minimizing discretization loss.

The results for other training contexts within NAS-Bench-201 are similar, though DARTS does not always actually attain the lowest entropy. However, the in DARTS-space operation bias becomes more complex as the significance of skip connections changes in a larger architecture. This is discussed further in Appendix F.

## 5.4 What aspects of the training context matter?

By comparing the overall ranking of algorithm combinations across each training context, we can observe the functional similarity of the problems these training contexts represent, with respect to our NAS algorithms. In order to do this, we group together the studied training contexts based on whether or not they share each of the three studied aspects of the training context: the number of epochs the supernet is trained for, the data set used to train the supernet and select and train the final architectures, and the search space the architecture is selected from. One example of this is that training contexts that differ only in their data set (while keeping the same search space and training length) retain more consistent ranking of combined NAS algorithms (median Spearman correlation of 0.68). Contexts that differ in their training length (and keep the same dataset and search space) show less consistent rankings of NAS methods (median Spearman correlation of 0.42).

Overall, the search space seems to be the defining component of the training context. The rankings of combined algorithms across training contexts with the same training epochs and data set but different search spaces had a median correlation of 0.08 while training contexts with the same search space and different training epochs and data sets produced a median correlation of 0.4. This suggests that evaluating NAS algorithms across different search spaces should be a significantly higher priority than evaluating across different data sets, and a fair comparison between algorithms

requires appropriate training length. Additional contextual comparisons can be found in Appendix F.

## 6 Conclusion

We address the lack of broad consideration of the problem of architecture selection for supernet NAS methods. We utilized a case study of two variations of DARTS to demonstrate that drawing generalizable insights about architecture selection requires more than proving the limits of an existing technique and proposing an alternative for a specific supernet training algorithm. Instead, we must begin by considering a wide range of supernet training methods in combination with architecture selection techniques to understand how these components of the NAS algorithm interact.

By measuring the performance of various supernet training and architecture selection methods in combination, we confirmed that the default and widespread architecture selection method of pruning is inferior to alternative algorithms in many training contexts. By comparing algorithms meant to address two competing explanations for this phenomenon, DARTS' overvaluing of skip connections and the sharp jump from continuous weights to discrete architecture, we weighed their relative contribution to DARTS' failure, showing that correcting the weighting of skip connections alone is not sufficient to surpass alternative architecture selection methods while sampling is. Finally, we analyzed broad trends across our combined evaluations to understand the components of the training context that determine the relative performance of NAS algorithms. Our results show that the search space used is the most significant factor, with the amount of epochs the supernet is trained for also playing a more major role in determining relative performance than the data set used. These results stress the importance of systematic evaluations of NAS algorithms using equivalent amounts of training for the compared algorithms, and show that evaluating across multiple search spaces should take higher priority than evaluation across multiple (similar) data sets. We hope these insights will help the research community evaluate and design the next generation of NAS algorithms.

## 7 Limitations and Broader Impact Statement

While our framing architecture selection positions it as a process which occurs after supernet training, this does not reflect how every supernet NAS algorithm operates, with methods like SGAS (Li et al., 2020b), which pruning or partitioning the search space during supernet training. Architecture selection is still being performed in these methods, but as it occurs iteratively, alternating with training, direct comparison to the methods reviewed here is not feasible. Extending the analysis of this work to these iterative architecture selection algorithms is, however, a promising direction for future work. A second limitation of this work is its focus on NAS-Bench-201. Although we do include the DARTS search space in our combined evaluations, extending this study to include additional and larger tabular benchmarks could help ensure robust and generalizable analyses. This is another opportunity for future work to expand upon the presented results.

This work serves to highlight and evaluate an understudied component of the supernet NAS process, with a core goal of advancing our scientific understanding of NAS. We intend to facilitate broader understanding of supernet NAS algorithms and provide guidelines for contributing to a general understanding of supernet NAS. Increased access to efficient NAS methods has the potential to lead to more widespread and more effective use of specialized deep learning models in science and industry. Of course, the proliferation of NAS methods does incur the general risks associated with widespread access to deep learning, including malicious or biased models. Additionally, we advance a unified understanding of zero-shot and one-shot NAS methods which we hope will promote the cross-study and use of energy efficient NAS algorithms with supernet training. Due to their efficiency of computation, zero-shot metrics can easily be used to provide a structural "sanity check" for selected architectures.

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

# Appendices

## A  Implementation Details

We trained each model on a single Tesla V100 GPU through an internal compute cluster. We utilized the random seeds [1, 2, 3] in our case study and [10, 11, 12] in our main experiment. The broad comparative experiments shown here required a total of 25400 GPU hours to compute and a trivial amount of CPU hours (<5) required for plotting. The source code utilized in the course of this experimentation is provided in full at: https://anon-github.automl.cc/r/arch_selection-6E4C/

Hyperparameters (other than the number of epochs the supernet is trained for) were fixed across all training contexts within a search space, but differed between NAS-Bench-201 (Dong and Yang, 2020) and the DARTS search space (Liu et al., 2019). We did not perform hyperparameter tuning, instead fixing the values at the defaults established for DrNAS (Chen et al., 2021b). This meant the supernet's shared weights were trained using SGD with a learning rate of 0.025, momentum of 0.9, and weight decay of 0.0003. The architecture weights were optimized using Adam, with a learning rate of 0.0003, weight decay of 0.001, and betas set at 0.5 and 0.999. The hyperparameters used for the DARTS-space experiments are as above, but with a learning rate of 0.1 for the shared weights and 0.0006 for the architecture weights. Also, in the DARTS search space we employ partial channel training, as developed by Xu et al. (2020), with k=2 across all algorithms.

As the set of supernet training algorithms in our study all train the supernet the amounts of computational time required for each all fall within the same scale, but GDAS is significantly faster than the other algorithms. For an example, we provide the mean run times for each algorithm run for 250 epochs on CIFAR in the NAS-Bench-201 search space in Table ??.

| Algorithm | Mean Run Time (GPU Hours) |
|---|---|
| DARTS | 9.95 |
| DARTS- | 10.32 |
| SNAS | 10.34 |
| GDAS | 5.74 |
| DrNAS | 10.15 |
| Perturb | 3.45 |
| All sampling | 2.50 |
| All zero-shot | 0.36 |

Table 3: Mean Run Times for independent search stages on CIFAR10 in NAS-Bench-201

The bottom section of Table ?? shows the computational time required for our architecture selection algorithms run on the "informed" baseline used for architecture selection evaluation. Perturbation-based selection was by far the slowest. In our experiments we iterated through the sampled architectures and computed both the validation accuracy via the supernet and the zero-shot metrics for each architecture and so cannot easily distinguish the time spent computing each measure from our records. In order to offer insight on the relative speed of these algorithms,

we present the total time used to compute all sampling-based architecture selection algorithms and then compute the zero-shot measures separately and present the total time used to compute just the zero-shot measures. This reveals that selecting based on validation accuracy is the second slowest studied architecture selection algorithm by a significant margin, taking roughly 2 hours for this baseline on CIFAR10. Note that the run time for perturbation-based selection scales with both the size of the search space and the size of the data set as it performs a number of additional epochs of fine-tuning based on the number of architecture parameters in the search space. Computing the validation accuracy does scale with both of these variables as well, but increases much less sharply than perturbation as the search space grows because the number of forward passes through the model is fixed with respect to the search space size. Therefore, in the larger DARTS search space, the difference in run time between these methods is exacerbated. The zero-shot methods are by far the fastest to compute, the recorded run time reflects the time necessary to compute not only the "synflow" and "jacob" measures, but also 6 additional measures not used in this study.

## B  Asset Information

| Asset | License | Link |
|---|---|---|
| DrNAS source code | Not specified | https://github.com/xiangning-chen/DrNAS |
| DARTS-PT source code | Apache 2.0 | https://github.com/ruocwang/darts-pt |
| Zero-Cost-NAS source code | Apache 2.0 | https://github.com/SamsungLabs/zero-cost-nas |

Table 4: Code assets used & licenses

The source code used for the experiments in this work incorporated portions of code from each of the above prior works whose algorithms are evaluated in the course of this study. In addition to the assets utilized in the programming of the experiment, this project makes use of the NAS-Bench-201 arhcitecture benchmark data and API, provided under an MIT license. All neural networks used in this work are implemented in PyTorch, which is released via an open source license in which each contributor retains the copyright for their contributions. Lastly, our neural architectures were trained using benchmark image data sets, CIFAR-10, CIFAR-100, and a downsampled version of ImageNet distributed with NASBench-201. The images in these data sets were obtained from the web and their copyright is unknown and not held by the researches who synthesized the data sets.

## C  Search space details

This work primarily focuses on two search spaces, that of NAS-Bench-201 and that of DARTS.

**NAS-Bench-201** The cells in the NAS-Bench-201 search space contain 4 nodes, including an input node, resulting in a total of 6 possible edges. One operation is selected for each of these edges out of 5 possible operations, consisting of a zero operation, a skip connection, 1-by-1 convolution, 3-by-3 convolution, and 3-by-3 average pooling. These cells are joined in 3 stacks of 5, separated by downsampling residual blocks, to form the full architecture as detailed in Dong and Yang (2020).

**DARTS** The search space used in DARTS is a variant of that proposed for NASNet (Zoph et al., 2018). The convolutional cells in this search space consist of seven nodes, including two input nodes, resulting in 14 edges with possible operations. In this search space, the outputs of the previous two layers are passed to the two input nodes and the output node concatenates the output of the other 4 non-input nodes without performing any additional operations. Each intermediate node is required to have two incoming edges, so exactly 8 out of the possible 14 edges with possible operations will be selected for each candidate architecture. Each of these edges may represent a 3-by-3 or 5-by-5 separable convolution, a 3-by-3 or 5-by-5 dilated separable convolution, 3-by-3 max pooling, 3-by-3 average pooling, a skip connection, or the zero operation. Two separate architectures fitting this

cell search space are learned for each model, a normal cell and a reduction cell, which is inserted 1/3 and 2/3 of the way through the stacked cells to perform spatial downsampling. The number of normal cells stacked varies between search and evaluation as detailed in Liu et al. (2019).

Additionally, in our case study, we utilize 4 constrained variants of DARTS' search space as developed by Zela et al. (2020).

- **S1** Each of the 14 possible edges in the DARTS search space has its own set of 2 possible operators, determined to be the most important two operators for that edge in the original DARTS search space.

- **S2** Like DARTS, but the set of candidate operations is restricted to 3-by-3 separable convolutions and skip connections.

- **S3** Like DARTS, but the set of candidate operations is restricted to 3-by-3 separable convolutions, skip connections, and the zero operation.

- **S4** Like DARTS, but the set of candidate operations is only 3-by-3 separable convolutions or "noise," where "noise" outputs a random feature map sampled from a standard normal distribution.

Note that the search space S4 is the only one which does not include skip connections. In Section 3, S4 was the only search space for which perturbation on a supernet trained without architecture weights was the clear best architecture selection technique for both DARTS and DARTS-. As DARTS- differs from DARTS only through adding an auxiliary skip connection to mitigate the overweighting of the skip connection in DARTS, it is unlikely that DARTS- would have been a helpful modification for a search space without skip connections.

### C.1 Selection of search spaces

We based the core of our study around NAS-Bench-201 due to its widespread use in NAS literature and its suitability for supernet NAS, as the definition of operations on edges, with up to one operation allowed on every possible edge means that any architecture weighting over edges can be simply discretized to a valid architecture. However, as this search space includes only 15625 possible architecture encodings, we decided to extend our evaluation to the DARTS search space. This search space is much larger (at $10^8$ architectures) and is the most common non-benchmark search space used in the supernet NAS literature.

## D  Review of architecture selection methods in prior work

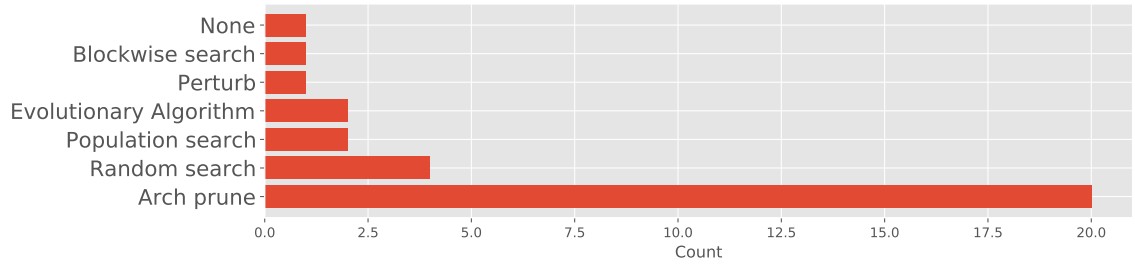

Figure 3: Architecture selection methods of published NAS algorithms

We review a set of 31 published supernet NAS algorithms to determine the nature of the architecture selection method used (Fig. 3). 28 of these algorithms above the horizontal line in Table 5 constitute every supernet-based NAS algorithm mentioned in Ren et al. (2021) (which is

inclusive of the supernet algorithms mentioned by Elsken et al. (2019)). The additional 3 algorithms are more recently published methods which we use as one of the supernet training methods in our investigation. We find that existing techniques overwhelmingly utilize some form of pruning the architecture weights or distribution to select a final architecture.

| Algorithm | S2 Search method | Notes |
|---|---|---|
| CNF (Saxena and Verbeek, 2016) | Arch prune | No arch weights, prune on op weights |
| DAS (Shin et al., 2018) | Arch prune | No topo search, progr fixed thresh prune |
| ENAS (Pham et al., 2018) | Random search | Uses trained sampler |
| MaskCn (Ahmed and Torresani, 2018) | Arch prune | Only learn inter-module connections |
| NAOnet (Luo et al., 2018) | Population search | Gradient based updates to population |
| OSNAS (Bender et al., 2018) | Random search | |
| BayesNAS (Zhou et al., 2019) | Arch prune | Entropy-based prune |
| DARTS (Liu et al., 2019) | Arch prune | Suggests multiple trials for final arch |
| DATA (Chang et al., 2019) | Arch prune | Variable amount of pruning |
| FBnet (Wu et al., 2019) | None | Randomly samples a subnet |
| GDAS (Dong and Yang, 2019c) | Arch prune | |
| I-DARTS (Jiang et al., 2019) | Arch prune | |
| PARSEC (Casale et al., 2019) | Arch prune | Sample during training |
| P-DARTS (Chen et al., 2019) | Arch prune | Progressive prune, alter search space |
| ProxylessNAS (Cai et al., 2019) | Arch prune | Sample during training |
| RSPS (Li and Talwalkar, 2019) | Random search | |
| SETN (Dong and Yang, 2019b) | Random search | Uses trained sampler |
| SNAS (Xie et al., 2019) | Arch prune | Prune ops, no method for edge pruning |
| TAS (Dong and Yang, 2019a) | Arch prune | No topology search |
| XNAS (Nayman et al., 2019) | Arch prune | Progr prune based on loss gradient bound |
| DNA (Li et al., 2020a) | Blockwise search | |
| FNA (Fang et al., 2020) | Arch prune | |
| PC-DARTS (Xu et al., 2020) | Arch prune | Combined prune on op, edge weights |
| PC-NAS (Li et al., 2020c) | Population search | Gradually grow candidate pool |
| R-DARTS (Zela et al., 2020) | Arch prune | Final arch selected from multiple trials |
| SGAS (Li et al., 2020b) | Arch prune | Progressive prune on multiple metrics |
| SPOS (Guo et al., 2020) | Evolutionary alg | |
| FairNAS (Chu et al., 2021b) | Evolutionary alg | |
| DARTS- (Chu et al., 2021a) | Arch prune | |
| DrNAS (Chen et al., 2021b) | Arch prune | |
| DARTS-PT (Wang et al., 2021) | Perturbation | |

Table 5: Detailed account of Architecture selection method review.

Table 5 provides greater detail on the distribution of Stage-2 search methods. As clarified in the notes, there is significant variation among the architecture techniques labeled as "arch prune." Many methods utilize more complex parameterizations of the architecture space than DARTS, but largely still prune based on magnitude. BayesNAS (Zhou et al., 2019) instead utilizes the entropy of the architecture parameterization as a pruning criterion, while SGAS (Li et al., 2020b) develops two different criteria incorporating magnitude, entropy, as well as the histogram intersection of the architecture parameterization over several updates.

This review of supernet NAS methods reveals two distinct strains of research. The dominant line of work investigates DARTS-like methods, primarily focusing on developing better methods of learning architecture weights and largely ignoring the possibility of alternative methods to

pruning for architecture selection. The other strain of supernet NAS research is concerned with "true" one-shot methods, those that, following the lead of Bender et al. (2018), do not evaluate subnetworks at all during the process of training the supernet. These algorithms use a random search, evolutionary algorithm, or, in the case of Li et al. (2020a), an iterative search algorithm which partitions the search space by the stacked blocks of the model, for Stage-2 search. The method of Wu et al. (2019) is unusual in that they learn architecture weights, and then simply sample a final architecture instead of pruning or sampling a pool of candidate architectures and estimating their performance (random search). In this work, we reconcile these two strains of supernet NAS research by demonstrating the utility of non-pruning architecture selection for DARTS-like supernet training algorithms and the value of biased supernet training for performance estimation based selection algorithms, like random search or evolutionary algorithms.

Also note that when "sample during training" is included in the notes, this signifies that discrete architectures are sampled during the training process, while the final architecture selection method does not depend on sampling. This is technically not true for GDAS (Dong and Yang, 2019c), as it samples a single operation on each edge in same search space utilized by DARTS which does not allow an operation on every edge, resulting in sampling a subset of the supernet that is not necessarily equivalent to any architecture within the search space. SNAS and DrNAS also utilize sampling during training, but use relaxed samples, not discrete architectures.

SGAS does utilize magnitude-based pruning, but not only once at the end of supernet training as originally performed by DARTS (Li et al., 2020b). ASAP combines this pruning-during-training with architecture weight annealing as used by SNAS and DrNAS resulting in an algorithm with PAC guarantees (Noy et al., 2020). These algorithms do not fit neatly into the simple sequential two-stage model presented in the main text. However, these same two stages can be extended to an iterative model of the NAS process in order to describe algorithms that prune during training..

## E  Further Evaluation of architecture selection

We seek to evaluate the ability of an algorithm to select the top discrete architectures from a given supernet. In order to evaluate the architecture selection process, we define a standardized input for the architecture selection search algorithms. We wish to provide a sufficiently broad baseline such that it is suitable for evaluating any architecture selection method which can be implemented on top of a supernet training algorithm. As we consider the supernet itself as the input to the architecture selection algorithms, our baseline must consist of a standardized set of supernets, trained in a manner such that evaluating architecture selection methods on them facilitates estimation of their efficacy across a range of supernet training methods.

By combining our training (or lack thereof) process with different methods of sampling candidate architectures, we synthesize four baselines from a single trained model. These baselines are as follows:

- *Untrained Supernet/Uniform Sampler* A supernet whose shared weights have been randomly initialized but not trained. For selection algorithms which require a sample of architectures, a uniform distribution over the search space is used.

- *Untrained Supernet/Informed Sampler* A supernet whose shared weights have been randomly initialized but not trained. For selection algorithms which require a sample of architectures, a distribution biased toward higher performing architectures is used.

- *Informed Supernet/Uniform Sampler* A supernet whose shared weights have been trained through sampling individual architectures. For selection algorithms which require a sample of architectures, a uniform distribution over the search space is used.

- *Informed Supernet/Informed Sampler* A supernet whose shared weights have been trained through sampling individual architectures. For selection algorithms which require a sample of architectures, a distribution biased toward higher performing architectures is used.

To be specific, the biased distribution described above as "informed" is a discrete probability distribution defined over the search space of NAS-Bench-201 where the probability of sampling an architecture is inversely proportional to its rank by test accuracy. This biased distribution is also the sampler used to train the randomly trained models. This sampler is designed to simulate a sampling based supernet training algorithm which is effective at identifying and sampling from high performing architectures during supernet training.

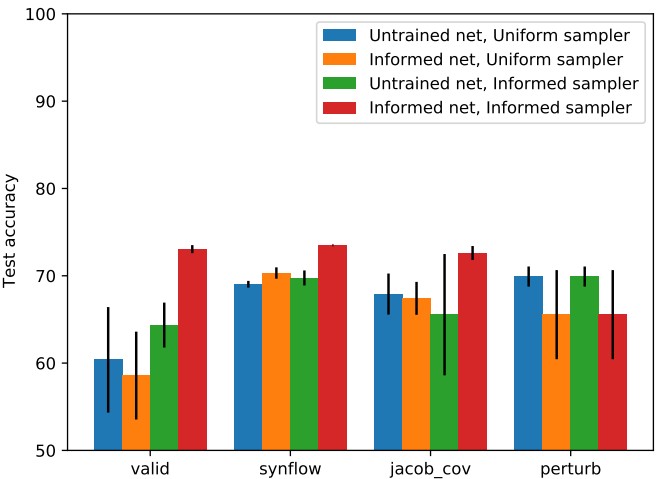

Figure 4: Comparison of Stage-2 search algorithm performance on CIFAR-100 with given baseline models. Baseline models are constructed by combining two components: a supernet, with shared weights which are either randomly initialized and *untrained* or have been training by sampling individual candidate networks using an *informed* sampler, and an architecture sampler, which uses either a *uniform* distribution or the *informed* sampler.

## F  Further Combined Evaluation

Tables 6 and 8 offer a summary of the trends discussed in our analysis of Stage-1 and Stage-2 combinations (Table 2). Viewing the algorithm combinations which produced the top architectures in Table 6 clarifies the dominance of SNAS and GDAS across Stage-1 algorithms and the variety of Stage-2 algorithms which offer effective alternatives to pruning in differing contexts. Viewing the counts of Stage-2 algorithms which outperform pruning in each context in Table 8, simplifies the trends in DARTS, DARTS-, and GDAS noted above. This display of the data also shows that SNAS is generally the Stage-1 algorithm best suited to pruning. Additionally, DrNAS, for which the original implementation was only trained for 50 epochs, like DARTS, seems to display the reverse of the pattern noted for GDAS. Pruning is an inferior Stage-2 algorithm for DrNAS when the supernet is trained too long, meaning 250 epochs on NASBench-201 or 100 epochs in the DARTS search space.

Fig. 5 offers a deeper look into the analysis of which aspects of the training context are most influential in determining the relative performance of algorithm combinations. The results showed more consistency in the combination rankings across different data sets with the supernets trained

| Search space | Data set | Epochs | Test Err | Stage 1 | Stage 2 |
|---|---|---|---|---|---|
| NASBench-201 | CIFAR10 | 50 | 5.66 | dirichlet | perturb |
| | | 100 | 5.64 | dirichlet | prune |
| | | 250 | 5.80 | dirichlet | synflow |
| | CIFAR100 | 50 | 27.23 | snas | prune |
| | | 100 | 27.48 | dirichlet | synflow |
| | | 250 | 28.43 | snas | synflow |
| | ImageNet-16-120 | 100 | 53.44 | snas | valid |
| DARTS-space | CIFAR10 | 50 | 2.65 | dirichlet | prune |
| | | 100 | 2.60 | dirichlet | prune |
| | CIFAR100 | 50 | 16.24 | snas | prune |
| | | 100 | 16.69 | gdas | prune |

Table 6: Top architectures in each training context studied.

| Search space | Data set | Epochs | DARTS | DARTS- | SNAS | GDAS | DrNAS |
|---|---|---|---|---|---|---|---|
| NASBench-201 | CIFAR10 | 50 | 4 | 1 | 0 | 4 | 1 |
| | | 100 | 4 | 2 | 0 | 3 | 0 |
| | | 250 | 4 | 3 | 0 | 0 | 4 |
| | CIFAR100 | 50 | 4 | 3 | 0 | 4 | 0 |
| | | 100 | 4 | 2 | 0 | 4 | 1 |
| | | 250 | 3 | 3 | 1 | 0 | 3 |
| | ImageNet-16-120 | 100 | 4 | 3 | 1 | 1 | 1 |
| DARTS-space | CIFAR10 | 50 | 2 | 0 | 1 | 1 | 0 |
| | | 100 | 2 | 1 | 1 | 0 | 0 |
| | CIFAR100 | 50 | 0 | 2 | 0 | 0 | 0 |
| | | 100 | 1 | 4 | 3 | 0 | 4 |

Table 7: Number of Stage-2 algorithms (out of 4) which surpassed magnitude-based pruning for each Stage-1 search algorithm in each context.

| Search space | Data set | Epochs | DARTS | DARTS- | SNAS | GDAS | DrNAS |
|---|---|---|---|---|---|---|---|
| NASBench-201 | CIFAR10 | 50 | 4 | 0 | 0 | 4 | 0 |
| | | 100 | 4 | 1 | 0 | 0 | 0 |
| | | 250 | 3 | 0 | 0 | 0 | 2 |
| | CIFAR100 | 50 | 4 | 3 | 0 | 4 | 0 |
| | | 100 | 4 | 0 | 0 | 4 | 0 |
| | | 250 | 1 | 0 | 0 | 0 | 3 |
| | ImageNet-16-120 | 100 | 3 | 3 | 1 | 0 | 1 |
| DARTS-space | CIFAR10 | 50 | 0 | 0 | 0 | 0 | 0 |
| | | 100 | 0 | 0 | 1 | 0 | 0 |
| | CIFAR100 | 50 | 0 | 0 | 0 | 0 | 0 |
| | | 100 | 0 | 1 | 0 | 0 | 2 |

Table 8: Number of Stage-2 algorithms (out of 4) for which the mean error minus one standard deviation attained via magnitude based pruning exceeds the mean error plus one standard deviation attained by the alternative Stage-2 algorithm.

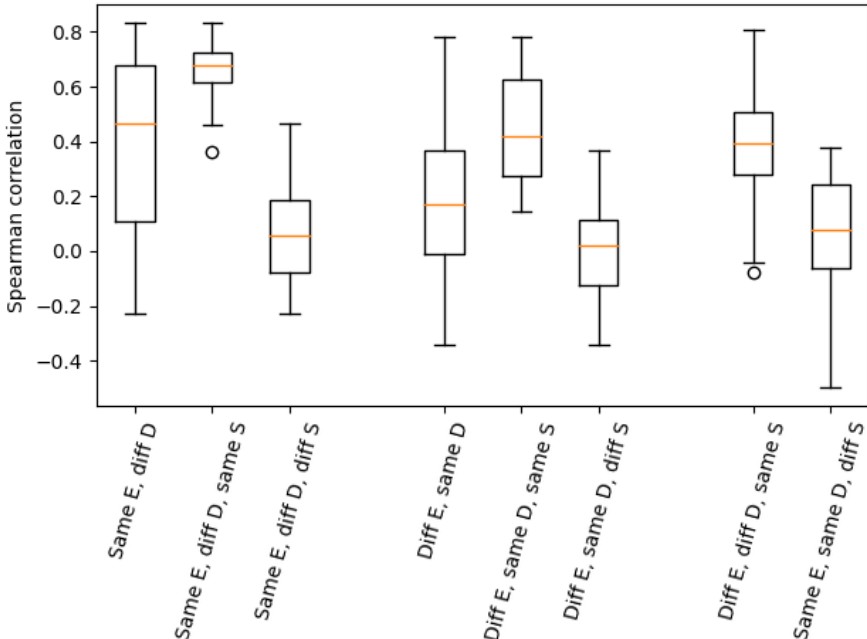

Figure 5: Distribution of Spearman correlation values for pairs of training contexts which satisfy the described grouping. The grouping label are abbreviated as follows: **E** is the number of epochs the supernet was trained for in Stage-1 Search, **D** is the image data set used during supernet training and architecture evaluation, and **S** is the architecture search space used (NASBench-210 or DARTS-space).

for the same number of epochs than when the supernets are trained for different numbers of epochs within the same data set. This may suggest that supernet training is more sensitive to internal dynamics of the training algorithm over time than it is to the specific image data used to train it. However, cross evaluation with other key training hyperparameters, like learning rate, would be necessary to explain precisely the relationship between epochs of supernet training and combined algorithm ranking.

For both cases of the same data set and same epochs of training, it's clear that there is little correlation in the algorithm rankings between NASBench-201 and the DARTS search space. As different encodings of different architecture spaces may result in vastly different gradient landscapes, it is unsurprising that rankings of optimization algorithms would be inconsistent across these differences. However, the result that varying everything but the search space leads to a median correlation of 0.37 while varying only the search space results in a median correlation of 0.11 is somewhat striking, underscoring the crucial importance of evaluating newly proposed NAS algorithms across multiple search spaces and cross evaluating competing NAS algorithms in the same search spaces.

For completeness, Tables 9 and 10 reports the standard deviation across the three runs of each Stage-1/Stage-2 combinations, as they were not able to be shown with the reported mean values in Table 2 due to space constraints. Note that the top value in each training context in general has relatively low standard deviation. This suggests that, despite the limited number of trials, our general observations about the superiority of particular algorithms or combinations will likely be consistent across additional trials. The high standard deviation values for GDAS with

| Data set | CIFAR10 | | | CIFAR100 | | | ImNet16 |
|---|---|---|---|---|---|---|---|
| Epochs | 50 | 100 | 250 | 50 | 100 | 250 | 100 |
| darts/prune | 45.7 ± 0.0 | 40.2 ± 7.8 | 15.8 ± 0.1 | 61.0 ± 0.0 | 61.0 ± 0.0 | 45.4 ± 0.0 | 81.6 ± 0.0 |
| darts/valid | 14.7 ± 8.1 | 9.1 ± 1.7 | 11.5 ± 3.2 | 51.0 ± 5.5 | 36.4 ± 3.2 | 47.5 ± 10.3 | 79.2 ± 5.4 |
| darts/synflow | 6.8 ± 0.2 | 9.5 ± 2.3 | 9.4 ± 1.8 | 30.1 ± 0.8 | 33.2 ± 0.5 | 42.8 ± 13.5 | 57.4 ± 0.9 |
| darts/jacob | 11.4 ± 1.2 | 10.0 ± 0.2 | 9.6 ± 1.1 | 33.2 ± 2.3 | 36.4 ± 1.2 | 35.9 ± 0.1 | 71.5 ± 4.5 |
| darts/perturb | 11.8 ± 0.1 | 14.3 ± 3.7 | 14.3 ± 3.6 | 40.4 ± 0.2 | 44.1 ± 5.6 | 43.0 ± 6.5 | 65.0 ± 1.2 |
| darts-/prune | 6.3 ± 0.1 | 9.4 ± 2.1 | 8.3 ± 2.5 | 32.8 ± 1.7 | 32.5 ± 4.9 | 32.1 ± 0.8 | 77.0 ± 4.7 |
| darts-/valid | 7.5 ± 0.5 | 11.9 ± 0.7 | 8.8 ± 2.5 | 38.2 ± 10.7 | 44.7 ± 7.5 | 34.0 ± 4.6 | 77.5 ± 5.2 |
| darts-/synflow | 6.8 ± 0.8 | 6.3 ± 0.5 | 6.3 ± 0.1 | 29.2 ± 1.6 | 28.5 ± 1.5 | 31.2 ± 2.7 | 54.7 ± 1.4 |
| darts-/jacob | 8.0 ± 0.6 | 8.2 ± 2.0 | 6.0 ± 0.3 | 29.0 ± 1.3 | 32.2 ± 2.5 | 31.1 ± 2.8 | 61.6 ± 5.1 |
| darts-/perturb | 6.2 ± 0.4 | 10.2 ± 1.2 | 8.1 ± 2.6 | 28.1 ± 1.0 | 35.1 ± 3.6 | 30.2 ± 2.0 | 64.7 ± 3.0 |
| snas/prune | 5.7 ± 0.1 | 5.9 ± 0.3 | 6.1 ± 0.3 | **27.2 ± 0.5** | 28.4 ± 1.3 | 30.4 ± 0.4 | 53.7 ± 0.0 |
| snas/valid | 14.8 ± 5.1 | 8.3 ± 3.4 | 7.5 ± 1.8 | 41.3 ± 10.9 | 37.0 ± 6.3 | 36.5 ± 9.4 | **53.4 ± 0.1** |
| snas/synflow | 6.6 ± 0.2 | 6.4 ± 0.1 | 6.2 ± 0.4 | 29.7 ± 1.1 | 29.1 ± 2.1 | **28.4 ± 1.9** | 56.9 ± 2.3 |
| snas/jacob | 7.9 ± 0.5 | 7.7 ± 0.3 | 7.7 ± 0.5 | 30.1 ± 0.6 | 32.5 ± 3.8 | 32.7 ± 0.1 | 54.6 ± 0.8 |
| snas/perturb | 6.2 ± 0.0 | 7.2 ± 1.5 | 7.3 ± 1.7 | 28.0 ± 0.2 | 33.7 ± 3.4 | 32.4 ± 0.5 | 56.2 ± 3.0 |
| gdas/prune | 34.8 ± 7.1 | 12.6 ± 8.5 | 6.6 ± 0.1 | 73.3 ± 12.7 | 83.6 ± 8.8 | 29.5 ± 0.5 | 59.0 ± 0.0 |
| gdas/valid | 21.5 ± 0.8 | 17.4 ± 4.4 | 9.4 ± 2.5 | 57.0 ± 1.3 | 53.0 ± 8.0 | 43.8 ± 1.7 | 69.3 ± 4.8 |
| gdas/synflow | 7.0 ± 0.2 | 7.0 ± 0.2 | 6.7 ± 0.1 | 30.8 ± 0.8 | 30.2 ± 0.6 | 30.5 ± 0.9 | 58.6 ± 0.8 |
| gdas/jacob | 7.5 ± 0.1 | 9.3 ± 1.3 | 8.5 ± 0.6 | 32.7 ± 2.4 | 36.2 ± 2.7 | 35.0 ± 0.5 | 63.4 ± 4.5 |
| gdas/perturb | 11.1 ± 1.1 | 8.3 ± 1.5 | 7.7 ± 0.5 | 38.4 ± 3.9 | 38.8 ± 4.1 | 31.4 ± 0.8 | 61.4 ± 4.8 |
| drnas/prune | 5.8 ± 0.3 | **5.6 ± 0.0** | 17.5 ± 8.8 | 28.0 ± 0.6 | 28.6 ± 1.6 | 45.4 ± 0.0 | 53.7 ± 0.0 |
| drnas/valid | 6.3 ± 0.1 | 6.4 ± 0.1 | 6.4 ± 0.2 | 32.4 ± 3.0 | 30.0 ± 0.6 | 53.4 ± 23.4 | 54.7 ± 0.2 |
| drnas/synflow | 6.4 ± 0.2 | 6.0 ± 0.3 | **5.8 ± 0.1** | 28.9 ± 0.1 | **27.5 ± 1.2** | 29.3 ± 2.4 | 53.7 ± 0.0 |
| drnas/jacob | 8.3 ± 0.6 | 7.4 ± 0.8 | 10.4 ± 0.5 | 32.9 ± 1.9 | 34.4 ± 3.8 | 34.7 ± 1.7 | 56.2 ± 1.5 |
| drnas/perturb | **5.7 ± 0.0** | 5.9 ± 0.2 | 11.1 ± 0.9 | 28.4 ± 0.7 | 29.3 ± 2.4 | 33.3 ± 0.0 | 56.0 ± 3.1 |

Table 9: The means with standard deviations corresponding to the mean values given in Table 2 for NAS-Bench-201. The underlined and bolded values corresponded to the underlined and bolded values in the mean table, such that bolded values are the lowest mean test error in that training context and underlined values are the combinations whose mean test error was below that of pruning for that Stage-1 algorithm in that training context.

lower numbers of epochs support our hypothesis that GDAS suffers from high uncertainty in its architecture weights without sufficient (longer) training.

### F.1 Operation Bias and Discretization Loss

As shown in Fig. 6, in the DARTS search space, we find that DARTS actually assigns the least weight to skip connections of the supernet training algorithms, while retaining the lowest entropy, despite not being the highest performing supernet training algorithm for any training context. The explanation for this phenomenon is multifaceted but can be summarized by noting that these measures are search-space specfic. First, as the DARTS search space consists of larger architectures and every node in the cell has two inputs, including skip connections can prove more helpful and is less likely to impede the inclusion of useful operations. This is made plain in Figure 7, which shows that the best discovered DARTS space architecture for CIFAR does include a skip connection and several highly effective architectures include multiple. Therefore, some bias towards the skip connection may actually be favorable in this search space and indeed, we see that, for CIFAR10, the top performing supernet training method, DrNAS, does assign relatively more weight to skip connections. Secondly, entropy can be misleading in this search space, as it is computed individually

| Data set | CIFAR10 | | CIFAR100 | |
|---|---|---|---|---|
| Epochs | 50 | 100 | 50 | 100 |
| darts/prune | 3.27 ± 0.44 | 3.08 ± 0.14 | 16.67 ± 0.36 | 17.03 ± 0.17 |
| darts/valid | 3.48 ± 0.47 | 3.16 ± 0.21 | 17.88 ± 0.42 | 18.60 ± 1.35 |
| darts/synflow | 3.66 ± 0.59 | 3.52 ± 0.08 | 17.64 ± 0.78 | 19.43 ± 0.58 |
| darts/jacob | 3.01 ± 0.23 | 3.04 ± 0.28 | 17.60 ± 1.23 | 17.56 ± 0.39 |
| darts/perturb | 3.25 ± 0.10 | 3.01 ± 0.06 | 17.50 ± 0.23 | 16.87 ± 0.38 |
| darts-/prune | 2.66 ± 0.15 | 2.89 ± 0.26 | 17.02 ± 0.42 | 19.89 ± 0.82 |
| darts-/valid | 2.98 ± 0.22 | 2.80 ± 0.17 | 17.09 ± 0.42 | 19.04 ± 1.14 |
| darts-/synflow | 3.22 ± 0.43 | 3.03 ± 0.07 | 17.48 ± 0.70 | 19.81 ± 1.75 |
| darts-/jacob | 3.11 ± 0.40 | 3.08 ± 0.37 | 16.98 ± 0.40 | 19.08 ± 1.00 |
| darts-/perturb | 2.72 ± 0.25 | 2.93 ± 0.17 | 16.92 ± 0.54 | 17.36 ± 0.25 |
| snas/prune | 3.10 ± 0.14 | 3.16 ± 0.07 | **16.24 ± 0.38** | 18.22 ± 1.45 |
| snas/valid | 3.19 ± 0.31 | 3.21 ± 0.31 | 16.79 ± 0.67 | 18.48 ± 1.21 |
| snas/synflow | 3.61 ± 0.67 | 3.23 ± 0.32 | 18.32 ± 0.63 | 18.19 ± 0.42 |
| snas/jacob | 3.13 ± 0.35 | 3.20 ± 0.21 | 16.41 ± 0.23 | 17.29 ± 0.31 |
| snas/perturb | 2.96 ± 0.32 | 2.96 ± 0.05 | 17.74 ± 0.46 | 16.96 ± 0.53 |
| gdas/prune | 3.04 ± 0.39 | 2.66 ± 0.09 | 16.49 ± 0.32 | **16.69 ± 0.69** |
| gdas/valid | 4.01 ± 0.62 | 3.55 ± 0.62 | 17.68 ± 0.91 | 18.49 ± 1.22 |
| gdas/synflow | 4.94 ± 0.29 | 4.24 ± 0.65 | 20.36 ± 1.18 | 20.31 ± 1.38 |
| gdas/jacob | 2.90 ± 0.11 | 3.30 ± 0.29 | 17.48 ± 0.75 | 17.21 ± 0.36 |
| drnas/prune | **2.65 ± 0.06** | **2.60 ± 0.09** | 16.73 ± 0.44 | 17.96 ± 0.38 |
| drnas/valid | 3.36 ± 0.07 | 2.74 ± 0.08 | 17.25 ± 0.79 | 17.55 ± 0.51 |
| drnas/synflow | 3.67 ± 0.66 | 3.46 ± 0.99 | 18.26 ± 0.67 | 17.06 ± 0.23 |
| drnas/jacob | 3.21 ± 0.32 | 2.84 ± 0.16 | 17.26 ± 0.05 | 17.12 ± 0.72 |
| drnas/perturb | 2.78 ± 0.05 | 2.77 ± 0.13 | 17.08 ± 0.62 | 17.10 ± 0.21 |

Table 10: The means with standard deviations corresponding to the mean values given in Table 2 for DARTS-space. The underlined and bolded values corresponded to the underlined and bolded values in the mean table, such that bolded values are the lowest mean test error in that training context and underlined values are the combinations whose mean test error was below that of pruning for that Stage-1 algorithm in that training context.

for each edge, however only 8 of the possible 14 edges are actually utilized by candidate architectures. The sampling based NAS algorithms are able to minimize discretization loss without near-discrete architecture weights by taking near-discrete (or discrete) samples from those architectures weights. The rest of the the entropy and skip connection weight plots are provided below for completeness.

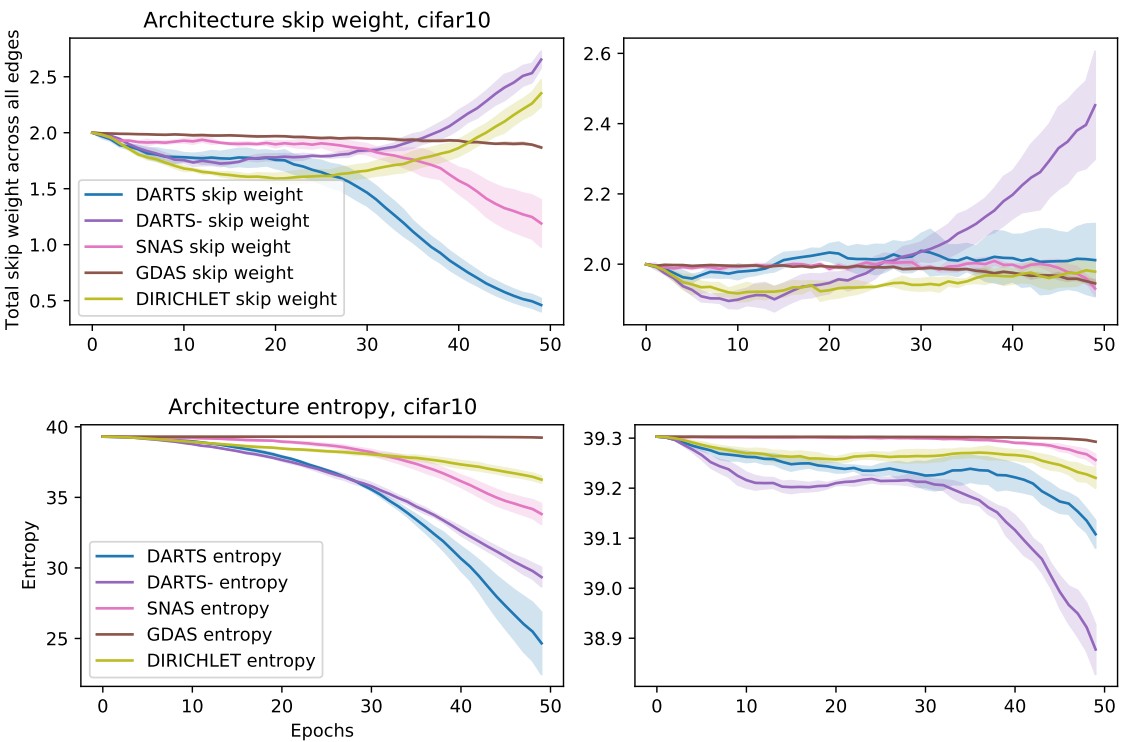

Figure 6: *Top:* The total weight (after softmax normalization) assigned to skip connections across all edges by each supernet training algorithm during training. *Bottom:* The entropy (summed over all edges) of the architecture weights for eah supernet training algorithm during training. Both plots show results for training with CIFAR 10 in the DARTS search space, with the results for the normal cell on the left and the results for the reduction cell on the right.

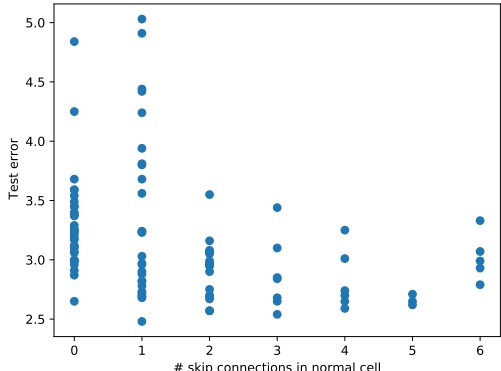

Figure 7: The accuracy and number of skip connections of all DARTS space final architectures trained on CIFAR-10.

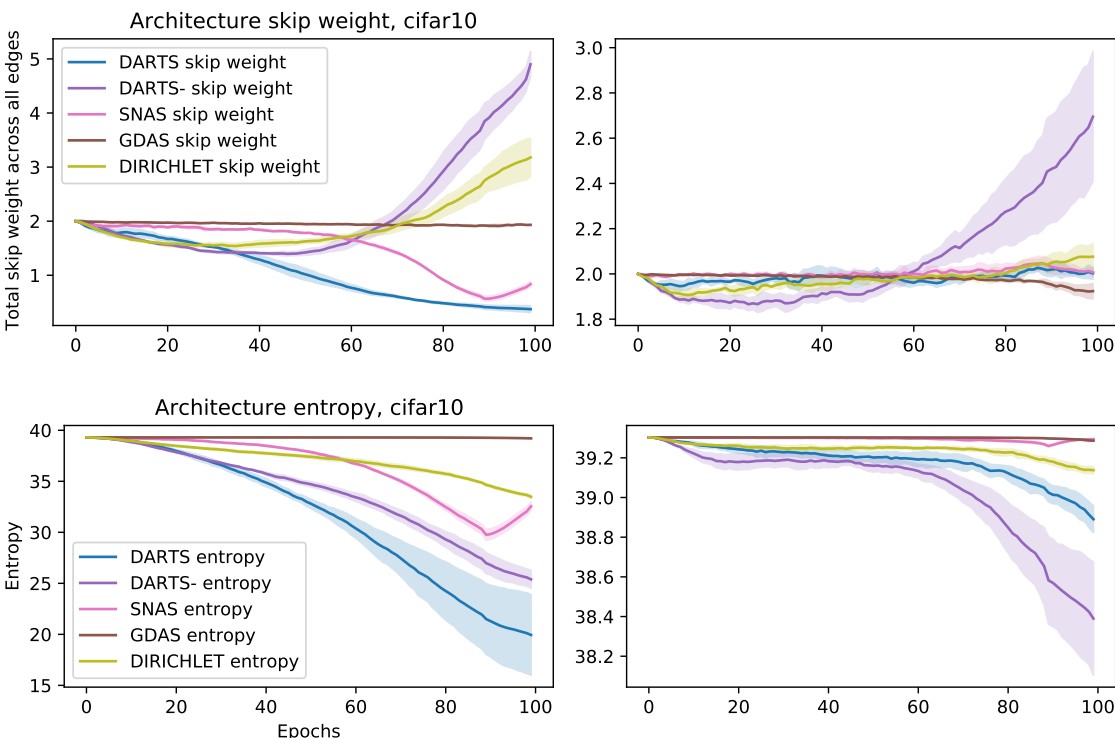

Figure 8: *Top:* The total weight (after softmax normalization) assigned to skip connections across all edges by each supernet training algorithm during training. *Bottom:* The entropy (summed over all edges) of the architecture weights for eah supernet training algorithm during training. Both plots show results for training with CIFAR 10 in the DARTS search space, with the results for the normal cell on the left and the results for the reduction cell on the right.

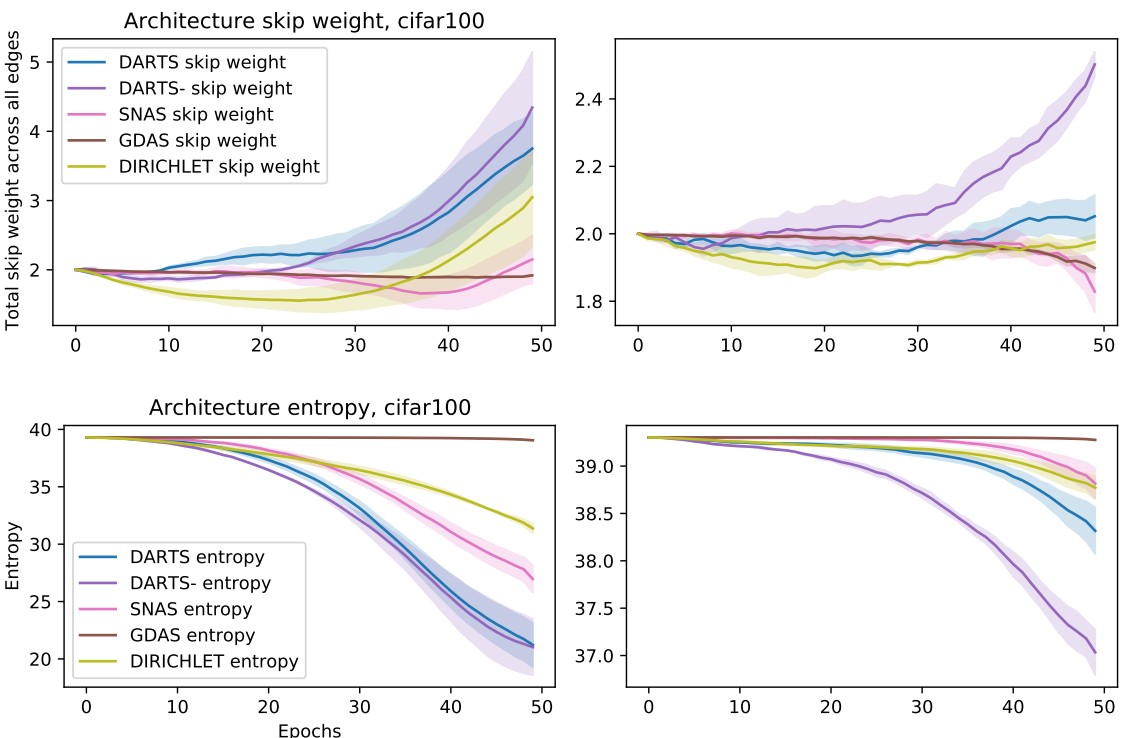

Figure 9: *Top:* The total weight (after softmax normalization) assigned to skip connections across all edges by each supernet training algorithm during training. *Bottom:* The entropy (summed over all edges) of the architecture weights for eah supernet training algorithm during training. Both plots show results for training with CIFAR 10 in the DARTS search space, with the results for the normal cell on the left and the results for the reduction cell on the right.

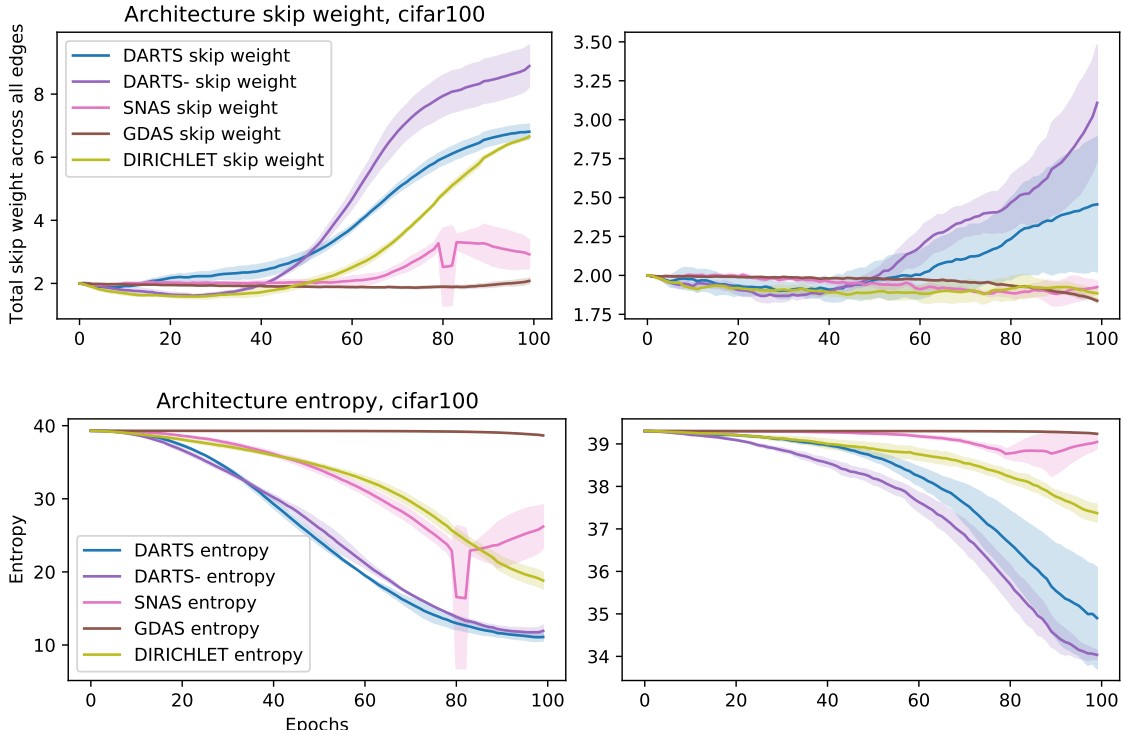

Figure 10: *Top:* The total weight (after softmax normalization) assigned to skip connections across all edges by each supernet training algorithm during training. *Bottom:* The entropy (summed over all edges) of the architecture weights for eah supernet training algorithm during training. Both plots show results for training with CIFAR 10 in the DARTS search space, with the results for the normal cell on the left and the results for the reduction cell on the right.

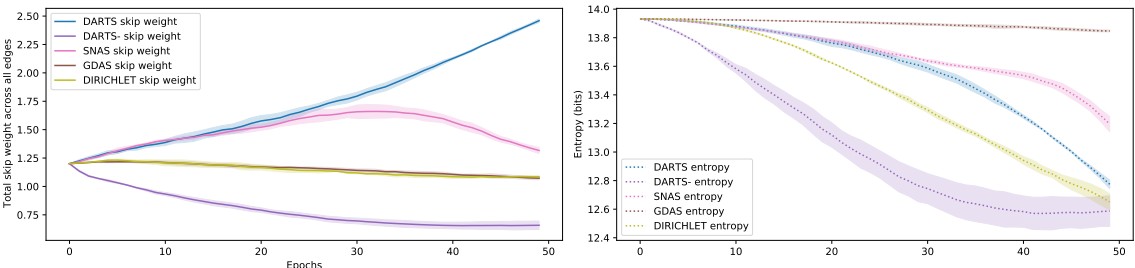

Figure 11: *Left:* The total weight (after softmax normalization) assigned to skip connections across all edges by each supernet training algorithm during training. *Right:* The entropy (summed over all edges) of the architecture weights for eah supernet training algorithm during training. Both plots show results for training with CIFAR 10 in the NAS-Bench-201 search space.

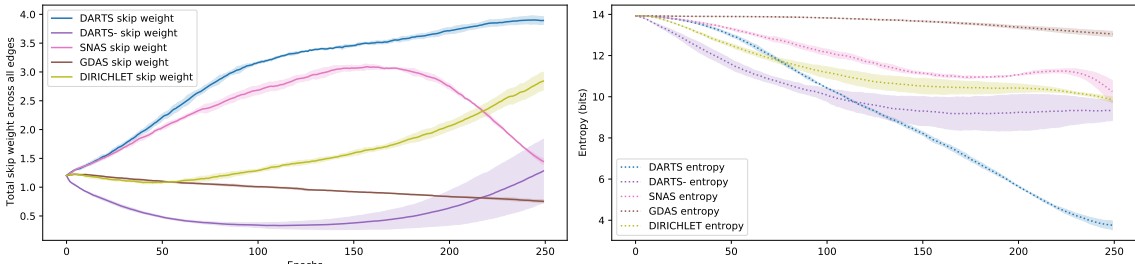

Figure 12: *Left:* The total weight (after softmax normalization) assigned to skip connections across all edges by each supernet training algorithm during training. *Right:* The entropy (summed over all edges) of the architecture weights for eah supernet training algorithm during training. Both plots show results for training with CIFAR 10 in the NAS-Bench-201 search space.

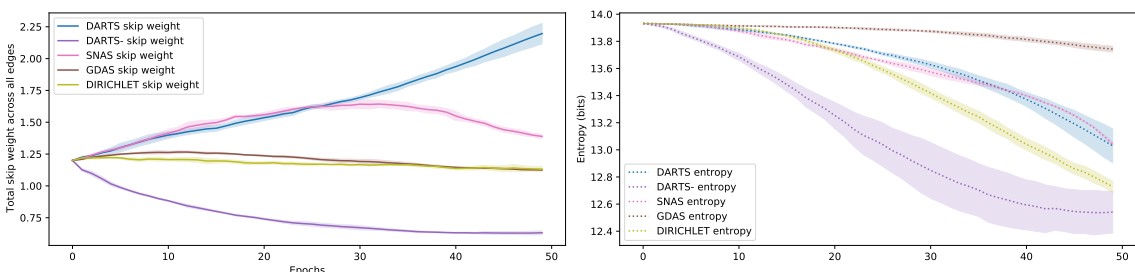

Figure 13: *Left:* The total weight (after softmax normalization) assigned to skip connections across all edges by each supernet training algorithm during training. *Right:* The entropy (summed over all edges) of the architecture weights for eah supernet training algorithm during training. Both plots show results for training with CIFAR 100 in the NAS-Bench-201 search space.

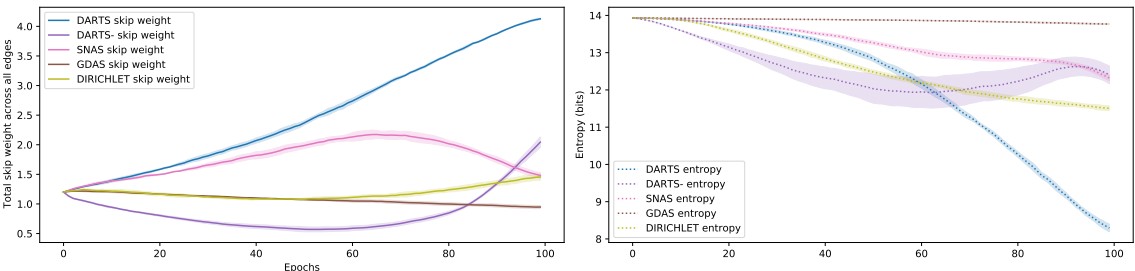

Figure 14: *Left:* The total weight (after softmax normalization) assigned to skip connections across all edges by each supernet training algorithm during training. *Right:* The entropy (summed over all edges) of the architecture weights for eah supernet training algorithm during training. Both plots show results for training with CIFAR 100 in the NAS-Bench-201 search space.

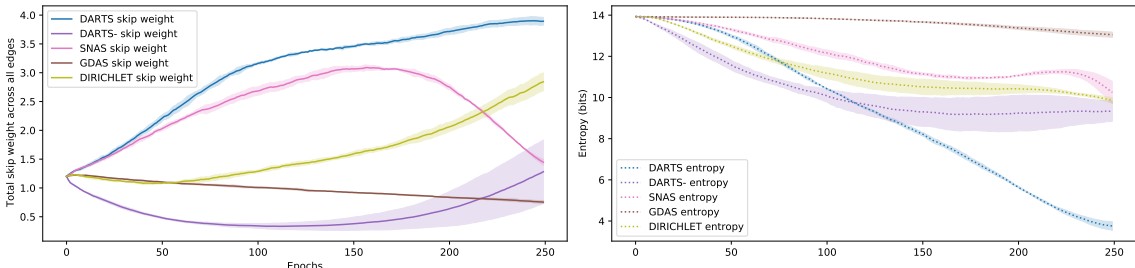

Figure 15: *Left:* The total weight (after softmax normalization) assigned to skip connections across all edges by each supernet training algorithm during training. *Right:* The entropy (summed over all edges) of the architecture weights for eah supernet training algorithm during training. Both plots show results for training with CIFAR 100 in the NAS-Bench-201 search space.

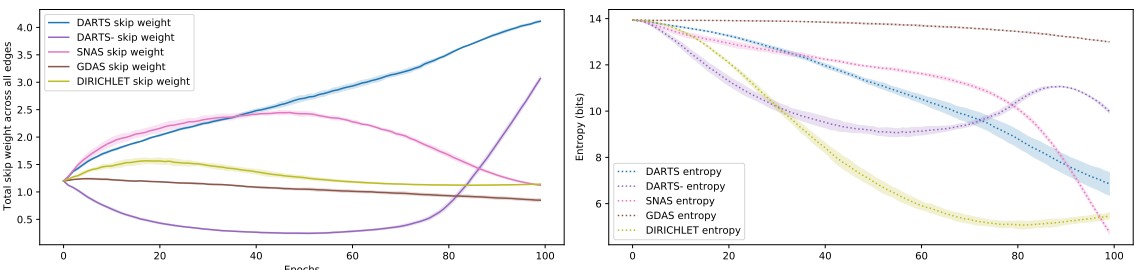

Figure 16: *Left:* The total weight (after softmax normalization) assigned to skip connections across all edges by each supernet training algorithm during training. *Right:* The entropy (summed over all edges) of the architecture weights for eah supernet training algorithm during training. Both plots show results for training with ImageNet16-120 in the NAS-Bench-201 search space.

