# OpenReview forum: "On the selection of neural architectures from a supernet"
_automl.cc/AutoML/2023/Conference — AutoML 2023 Workshop_

### Official Review · Reviewer_QfjP · 2023-04-10

**Potential Impact On The Field Of Automl Rating:** 1
**Technical Quality And Correctness Rating:** 2
**Clarity:** The paper is very well written
**Clarity Rating:** 4

**Summary Of Contributions:**

The paper presents a comprehensive assessment of architecture selection techniques along with various supernet training methods, highlighting the previously overlooked interdependence among them. It also showcases the effective integration of zero-shot NAS methods as new architecture selection mechanisms in supernet NAS training while evaluating the aforementioned techniques

**Actions Required To Increase Overall Recommendation:**

it is hard to improve the paper since it is an empirical study of a reasonably large search space. The lack of consistent outcomes and generic take home messages across setups makes it just not worth acceptance.

**Overall Review:**

I want to first of all highlight the effort for designing the framework by taking the search space, the datasets and the number of training epochs to evaluate different pairs of supernet training and architecture selection mechanisms. Running all possible combinations resulted in a very high computational cost (25400 GPU hours). This very high cost is the actual reason that pushes practitioners to focus on a single aspect which is mostly the supernet training. Obviously, they are aware of the issues related to the architecture selection mechanisms and the interplay between those and the supernet training strategies and not as claimed by the authors. The whole point of AutoML is not only to reduce the manual workload but also to keep the computational load under control. Apart from this major critisism, the take home-messages of the paper regarding the role of the search space, the training length and the dataset are not novel.

**Potential Impact On The Field Of Automl:**

The paper provides an empirical verification of common intuitions. The take-home messages are neither consistent across different setups nor novel.

**Review Confidence:**

5: You are absolutely certain about your assessment. You are very familiar with the related work and checked all the details carefully.

**Review Rating:**

3: Reject: For instance, a paper with technical flaws, weak impact, and/or weak evaluation.

**Review Summary:**

Given the lack of novelty and the minor expected impact on the field of AutoML, this paper is not eligible for acceptance at this venue.

**Technical Quality And Correctness:**

The paper builds strongly on empirical results and I have not identified any flaw with regard to the contributions. I want on the other side to highlight one major claim of this paper: "demonstrate the previously underappreciated interdependence between various methods for supernet training and architecture selection mechanisms". This is a controversial claim that is not well supported. More details will follow in the overall review section.

---

> ### Author Response · Authors · 2023-05-02
> **Response to Reviewer QfjP**
>
> Thank you for your review. We are glad that you found the empirical results to be strong and the paper clearly written.  In response to your comments:
>
> The take-home messages are neither consistent across different setups nor novel.
>
> * We agree that the initially submitted version of this paper did not clearly present the key-takeaways of our study. We have revised the paper to make the core message much more clear (including a bulleted list of contributions at the end of the Introduction) and better demonstrate the novelty of this finding compared to prior work.
>
> The paper provides an empirical verification of common intuitions.
>
> * We find value in empirical verification of common intuition, and view it as a hallmark of a sound scientific process.  Though we certainly admit that it is not as flashy as a surprising finding.
>
> The whole point of AutoML is not only to reduce the manual workload but also to keep the computational load under control
>
> * The intent of this work is to leverage toy problems and existing benchmarks to try and gain understanding of the relationship between supernet training and architecture selection.  While we hope that the understanding from this paper will help to inform future studies, we do not think it is necessary, nor wise, for future authors or practitioners of NAS to necessarily replicate the immense computational study we undertake here – though perhaps the lesson learned from this work might persuade them to present the results of a novel supernet training algorithm in the context of multiple architecture selection methods for a broader picture of its potential success (and maximum performance), as this is not commonly done in the field right now.
>
> I want on the other side to highlight one major claim of this paper: "demonstrate the previously underappreciated interdependence between various methods for supernet training and architecture selection mechanisms". This is a controversial claim that is not well supported.
>
> * We apologize for the poor phrasing, and are absolutely not suggesting that other NAS authors are ignorant to the potential issues with architecture selection or have to always publish broad comparative studies like this one. Our assertion was not that studies which focus on a single aspect of the NAS process in the course of presenting a new algorithm are flawed, but merely that broad comparative studies such as this one are also necessary to develop generalizable explanations of the NAS process, and are less representative of the currently published works in the field (perhaps in part due to your observation that unintuitive and concise take-home messages are not typically highly valued).

---

### Official Review · Reviewer_B96n · 2023-04-11

**Potential Impact On The Field Of Automl Rating:** 2
**Technical Quality And Correctness Rating:** 3
**Clarity Rating:** 3
**Actions Required To Increase Overall Recommendation:** 1. I would appreciate the authors ins…

**Summary Of Contributions:**

This paper broadly considers the task of architecture selection from supernet based methods for neural architecture search. The authors study DARTS- and DARTS -PT to claim that search space design is the most significant factor in NAS, with epochs required to train the supernet being more important than the data-set used. The paper makes a case for the importance of systemic evaluation of NAS algorithms.

**Clarity:**

I think the paper is presented in a clear way.

I do not believe Figure 4 is explained in the Appendix, I think it is useful, and adding an explanation to it would help.

**Overall Review:**

This is a timely, well motivated paper with interesting insights. The case study provides a clear understanding of the limitations of DARTS as well as the differences in methods of addressing the instability of architecture weights in DARTS. The results are well presented and conclusions are well supported by available evidence. However, the diversity of data-sets and search spaces brings into question the quality of available evidence. The main negative aspect of the paper is the lack of diversity in the architectural search space, making their comments on the importance and trade-offs of architecture design, architecture selection methods and training lengths inadequately supported.

**Potential Impact On The Field Of Automl:**

This is a well motivated paper, with a focus on studying the importance of architecture selection methods, architecture design methods and the training lengths. While the potential impact for the ideas behind the paper are high, the paper focuses on only two search spaces, NAS-Bench-201 and DARTS. In DARTS, 4 constrained versions of DARTS search space is utilized. While these search spaces are explained well, the potential impact of the insights in this paper are heavily limited by the lack of diversity in data-set and search spaces. Further details are in the technical quality and correctness.

**Review Confidence:**

4: You are confident in your assessment, but not absolutely certain. It is unlikely, but not impossible, that you did not understand some parts of the submission or that you are unfamiliar with some pieces of related work.

**Review Rating:**

5: Borderline Leaning Reject: Technically sound paper where reasons to reject nonetheless outweigh reasons to accept. Please use sparingly.

**Review Summary:**

I recommend a weak reject. I do not believe there are any technical flaws in the study, but its impact would be limited due to the weak evaluation. The paper addresses an important problem in differentiable NAS and conducts an effective literature survey to identify aspects of differentiable NAS that requires further study. However, a greater diversity of search spaces and data-sets are required to convince me of the conclusion and contributions.

**Technical Quality And Correctness:**

On the NASBench-201 data-set, CIFAR10, CIFAR100 and IN16 are highly correlated.

| Spearman Rank Correlation | CF10 | CF100 |
|---------------------------|------|-------|
| CF100                     | 0.97 | 1     |
| IN16                      | 0.95 | 0.96  |

A study regarding the architecture-data-set correlation within these search spaces and its consequences on the conclusions/contributions of the paper would help strengthen the main points of the paper.

Table 2 in the paper is extremely useful, as it focuses on a wide range of supernet training and architecture selection algorithm combinations. However, the insights are limited by limited diversity in search space and data-sets.

---

> ### Author Response · Authors · 2023-05-02
> **Response to Reviewer B96n**
>
> Thank you for your thoughtful review of our work. We appreciate your comments on the potential impacts of the ideas behind the paper and our evaluation of supernet training and architecture selection combinations in Table 2.  In response to your required actions to increase the overall recommendation:
>
> I would appreciate the authors insights on whether their selection of search spaces and data-sets are sufficient to support the claims made in the paper.
>
> * We have also added a section discussing our choice of search spaces to Appendix C. While NAS-Bench-201 does represent a fairly limited search space, we have also incorporated results from the search space used by DARTS (as well as NAS-Bench-301), which is a much larger search space. Given the observed significant variation in results across search spaces, it did not seem that presenting results in a third search space would clarify the core message of our work.
>
> Why are full-ImageNet scale evaluations not included in the paper? I believe they would make the paper more insightful.
>
> * Full-ImageNet results are not used for NAS-Bench-201 because the full version of the data set is not part of the benchmark. Full ImageNet results would certainly provide an interesting data point, and a scaled up version of this study would be a promising direction of future work. However, given that our results, detailed in Appendix F, indicate that the ranking of supernet training/architecture selection algorithm combinations varies more across different numbers of epochs of training than different image data sets, and that CIFAR 100 remains somewhat difficult for DARTS-space architectures, suggesting the difficulty of our existing settings.  Further confidence in these findings through other difficult and different datasets in the future will certainly add to the concepts highlighted here.

---

> > ### Comment · Reviewer_B96n · 2023-05-04
> > **Resposne To Authors**
> >
> > Thank you for your response.
> >
> > I understand that NASBench-201 does not have a full imagenet version of the data-set, but several existing papers on super-networks focus on full-scale ImageNet, going as far back as 2020 (DSNAS, SNAS etc.). However, I can understand that it is more reasonable to present results on CIFAR-10 and CIFAR-100, given that DARTS has a search space of 10^8 architectures.
> >
> > I find the idea of using zero cost methods for selecting architectures an interesting thought, and I am not surprised that it works. Whether the inherent FLOPs/Params bias is propagated by such zero cost selection methodologies is also an important question that I am not sure is evaluated here. Kindly correct me if I missed it.
> >
> > I completely agree with the authors remark on Line 331:
> > 'This suggests that evaluating NAS algorithms across different search spaces should be a significantly higher priority than evaluating across different data sets, and a fair comparison between algorithms'
> >
> > I think the ideas in the paper are interesting, but evaluating the proposed methods on more design spaces is central to such a paper which focuses on empirical evidence for the statements made. It may be a personal opinion, but I would suggest making Table 2 more succinct, and to separate out the conclusions drawn from epochs and search space (NASBench-201 and DARTS-space), as it is a bit difficult to register the underlined observations on the table.
> >
> > I have raised my score to 5, as I am still concerned about overall novelty and the number of search spaces the test is conducted on. I understand full-ImageNet may not be compulsory to support the statements made in the paper.

---

### Official Review · Reviewer_64dj · 2023-04-13

**Potential Impact On The Field Of Automl Rating:** 4
**Technical Quality And Correctness Rating:** 3
**Clarity Rating:** 3

**Summary Of Contributions:**

The authors of this paper conducted an analysis of NAS algorithms to better understand the conceptual structure of supernet NAS algorithms and their performance. They conducted a literature review and an experimental case study to highlight the need for architecture selection and supernet training to be studied as interdependent processes. They then performed comparative evaluations of these algorithms. The authors stress the importance of systematic evaluations of NAS algorithms using equivalent amounts of training for compared algorithms and across multiple search spaces. The insights gained from this study can guide the development of future differentiable search methods.

**Actions Required To Increase Overall Recommendation:**

As suggested before, the paper can be enhanced by conducting experiments on larger benchmarks and more architecture selection methods. The presented insights from the results on NAS-Bench-201 are confusing in a sense. It may be good separate zero-shot methods and give an in-depth analysis separately from the DARTS-based methods.


The following latest papers should be considered in references to support the literature review:
Weight-sharing neural architecture search: A battle to shrink the optimization gap
SuperNet in Neural Architecture Search: A Taxonomic Survey
Neural Architecture Search Survey: A Hardware Perspective
Neural Architecture Search: Insights from 1000 Papers

Minor issues:

Reference to Table 3 is missing in lines 548 and 549
Line 187: Here we focus in on the -> Here we focus on the


**Clarity:**

The analysis in the limited space is presented in a good way, and the meta-analysis is good for future researchers to understand the network selection methods.

**Overall Review:**

NAS-Bench-201 is small and inexpressive as it contains only 15k architectures. The results on this benchmark are not generalizable and may not transfer to larger benchmarks. Therefore, larger NAS Benchmarks should be considered for evaluation. Or else, the authors should give justification for choosing only this benchmark.

There are many similar works in the past targeting architecture selection methods. It is difficult to understand the differences with other papers. A comparison table in Section 2.3 (or in the appendix) could provide a better understanding of the main contribution of this paper over others.

It may be good to plot the searched models for a few cases and compare the architectural specifications for different methods.

The size of the sampled/searched model matters. So, a deeper analysis of how model size varies with the architecture selection method may be potentially useful along with accuracy.

The authors considered only prune, valid, synflow, jacob, perturb for evaluation. What is the reason for choosing only these five? Are these the only methods available?

This work says “We did these experiments and here are the results with the interpretation of results” It may be good to provide any further guidelines for future researchers to improve the selection methods.


**Potential Impact On The Field Of Automl:**

This paper can have great potential in the field of Differentiable Neural Architecture Search as the meta-analysis conducted in this paper provides new insights. It is not straightforward to choose the architecture selection from a trained super network as it depends on different factors.

**Review Confidence:**

4: You are confident in your assessment, but not absolutely certain. It is unlikely, but not impossible, that you did not understand some parts of the submission or that you are unfamiliar with some pieces of related work.

**Review Rating:**

6: Borderline Leaning Accept: Technically sound paper where reasons to accept outweigh reasons to reject. Please use sparingly.

**Review Summary:**

The reviewer mainly focused on interpreting the results of benchmarking different architecture selection methods. The reviewer has gone through the methods chosen as a part of the study and provided a few steps to improve the article.



**Technical Quality And Correctness:**

Weight-sharing/supernetwork-based methods are an important class of NAS algorithms, and the comparative analysis is important to understand to better understand how to sample the final network from the supernetwork. The detailed analysis of 5 architecture selection methods on different search methods is interesting. The analysis in Sections 4.1 and 5.1 are well written to better understand the results.

---

> ### Author Response · Authors · 2023-05-02
> **Response to Reviewer 64dj**
>
> Thank you for your highly informative and well researched review, your feedback is greatly appreciated. We are glad that, despite clarity issues in the original submission addressed in other reviews, you were able to find the core contributions of this paper and find them to be of great potential impact on the field.  In response to your required actions to increase the overall recommendation:
>
> The following latest papers should be considered in references to support the literature review: Weight-sharing neural architecture search: A battle to shrink the optimization gap SuperNet in Neural Architecture Search: A Taxonomic Survey Neural Architecture Search Survey: A Hardware Perspective Neural Architecture Search: Insights from 1000 Papers
>
> * We definitely agree that in the originally submitted version of this paper, we did not do a sufficient job distinguishing our work from past meta-analyses of NAS. We have since revised the writing to make our contributions more clear, denoted explicitly at the end of our introduction. We have additionally incorporated the perspectives of the suggested surveys, clear in our new discussion of operation bias and discretization loss.
>
> The paper can be enhanced by conducting experiments on larger benchmarks and more architecture selection methods. The presented insights from the results on NAS-Bench-201 are confusing in a sense. It may be good separate zero-shot methods and give an in-depth analysis separately from the DARTS-based methods
>
> * We certainly agree that NAS-Bench-201 is smaller in size and that generalizable conclusions require the use of larger search spaces, which is the reason that we also utilized the DARTS search space (which is the same search space of NAS-Bench-301) for our combined results. Based on the comparison of the NAS-Bench-201 results and DARTS results revealing very little consistency in rankings of architecture-selection/supernet-training algorithm combinations across search spaces, we believed adding an additional search space would require considerable additional computational expenditure and be unlikely to clarify the core results of this paper.  We have incorporated an additional section in Appendix C explaining our choice of NAS benchmarks.  However, we do think expanding this study to other search spaces, especially for models other than computer vision, is a promising direction for future work. As for our selection of architecture search methods, we have added additional discussion of this experimental design in Section 4. While the approaches we selected are not exhaustive of all published architecture selection approaches, we do believe that they meaningfully represent the range of adopted approaches.

---

> > ### Comment · Reviewer_64dj · 2023-05-07
> > **Reply to Authors**
> >
> > The authors did not provide a detailed response to many of the raised concerns or revised the manuscript based on the reviews provided. The paper still has flaws which are yet to be addressed. As suggested by other reviewers as well, the lack of diversity in terms of experiments and benchmarks makes it difficult to believe the true value of this work. The authors provided the results in a good way, however, the interpretation is confusing. Also, it is still unclear for future researchers to effectively utilize this study to design novel sampling and Supernetwork based methods. The reviewer is still skeptical about the limited study and difference between the previous meta-analysis works. However, this study is still important for the existing body of literature. Hence, the reviewer rating is still 6 (Borderline Leaning Accept).

---

### Review · Reproducibility_Reviewer_a4op · 2023-04-13

**Completeness Of Code And Dataset Supplement Rating:** 4
**Usability And Ease Of Reproducibility Rating:** 4

**Actions Required To Increase The Reproducibility And Overall Recommendation:**

I believe the reportability of the code is sufficient and does not require additional improvements.

**Completeness Of Code And Dataset Supplement:**

The supplemented code is complete. I was able to run the code and I believe the result are reducible by given code.

**Overall Reproducibility Review:**

The reproducibility of the work is very well. The code and appendix of the paper provide all necessary instruction to run the code. It clearly separates different experiments in paper and is straightforward to run.

**Review Confidence:**

4: You are confident in your assessment, but not absolutely certain. It is unlikely, but not impossible, that you did not understand some parts of the submission or that you are unfamiliar with some pieces of the code or data.

**Review Rating:**

10: Exceptional, this paper is reproducible at the push of a button and a model for the community.

**Review Summary:**

I was able to run the code with ease and the code is consistent with results in the paper.

**Summary Of Necessary Code And Dataset Supplement:**

Case study extends DARTS-PT (Wang et al., 2021) published code to implement DARTS- (Chu et al., 2021a).  Two Stage Supernet Search experiments extends published  code for DrNAS (Chen et al. 2021) to include  implementing DARTS- and RSPS in the NAS-Bench-201 search space. The baselines are implemented with random search with parameter sharing (RSPS). The Search spaces used are DARTS (Liu et al. 2018) and NASBench-201 1 (Dong and Yang, 2020), with CIFAR10, CIFAR100, IN16 datasets.

**Usability And Ease Of Reproducibility:**

The code was relatively easy to run, and the documentation is sufficient. I believe that the main results of the paper are easily reproducible with the code.

---

> ### Author Response · Authors · 2023-05-02
> **Response to Reproducibility Reviewer a4op**
>
> Thank you for your deep dive into our manuscript and codebase.  We are glad that you found the code easy to run, the results reproducible, and the documentation sufficient.  This is often thankless work, and we genuinely appreciate it.

---

### Official Review · Reviewer_FmZd · 2023-04-21

**Potential Impact On The Field Of Automl Rating:** 2
**Technical Quality And Correctness Rating:** 2
**Clarity Rating:** 3

**Summary Of Contributions:**

The paper is about methods for training supernets, ie network architecture selection. It is mostly a detailed comparison of existing methods across existing benchmarks, with key parameters varied in order to establish performance ranking of methods and relate this to the underlying setting. Some rather subtle findings emerge and there is a reasonable attempt to relate this to salient features of the methods compared.

**Actions Required To Increase Overall Recommendation:**

The authors should improve the abstract to deliver a clearer finding / improve the chances of impact.

The authors should more clearly present testable hypotheses, or if this is not possible, then suggest ways in which findings could be tested in future.

**Clarity:**

The overall approach does not come across brilliantly from the writing; either the study was not fully motivated and well-designed, or the writing did not clearly bring that across.
The abstract does not highlight a clear finding, and could probably be improved.
The background context and the experimental settings and results are clearly written. The paper is also well-structured.
So overall the paper is clear, with some reservations.

**Overall Review:**

The paper is a large and thorough comparative assessment of some techniques that certainly fall within AutoML.

The paper is somewhat justified by a discussion of the context of the methods and setting, which go some way to motivating the study.

Experiments are presented clearly and code is provided in support.

On the other hand,

The findings of the paper appear to be subtle.

The abstract does not highlight a clear finding.

The experiments would be best described as explorative, rather than designed carefully to assess a distinct hypothesis. This means that conclusions (where attempted) tend to be 'just-so', and are not then tested further. This makes the paper less valuable.

**Potential Impact On The Field Of Automl:**

The findings of the study are subtle and do not appear to be of high importance. I would not rule out the potential for citations of this work, but there is not a bold claim here that makes it obvious. On the other hand, the work seems to be thorough and fairly exhaustive. Other authors may find that the findings are useful (even if subtle) and are saved a lot of work by the existence of this large comparative assessment.

**Reproducibility (Optional):**

The paper appears to be reproducible in that links are provided to code. I have not checked this repository myself however.

**Review Confidence:**

2: You are willing to defend your assessment, but it is quite likely that you did not understand the central parts of the submission or that you are unfamiliar with some pieces of related work.

**Review Rating:**

4: Weak Reject: For instance, a paper with minor technical flaws, limited impact, and/or weak evaluation.

**Review Summary:**

My recommendation is based on decades of experience in ML and related areas, and an assessment largely of how clearly the authors have justified and presented their findings, and my estimation of how much interest/impact that will have. I accept that I could be wrong because this specific area of AutoML is not my area.

**Technical Quality And Correctness:**

This is a large study, and it appears correct, and replicable. This would tend to suggest a moderate rating at least.

However, although the experiments conducted are justified/motivated by a description of the context, specific hypotheses are not really raised and tested. Rather, experiments are performed, results are observed and some patterns from the results are derived by applying rank correlations. Then these patterns are somewhat justified by reference back to the features of the methods and details of the context tested in, but these 'theories' are then not tested again independently or in some other way. For this reason, I am not sure of the value and hence the real technical quality.

---

> ### Author Response · Authors · 2023-05-02
> **Response to Reviewer FmZd**
>
> Thank you for your thorough and informative review.  Your comments on how to maximize the impact of this paper are much appreciated.  We appreciate your positive view of our technical quality and have attempted to improve the clarity of the framing and, as you point out, thus also the potential impact of the manuscript.  In response to your required actions to increase the overall recommendation:
>
> The authors should improve the abstract to deliver a clearer finding / improve the chances of impact.
>
> * We agree that the previous abstract did not effectively highlight the contributions of this paper and we have revised the abstract, main text, and added a bulleted list of contributions to better highlight the noteworthy findings. We believe your assessment of the findings of our paper as subtle is accurate, there is not a single big splashy takeaway from this paper. However, the findings of this paper do address some key issues and ongoing debates within the NAS literature and we believe them very important to advancing our collective understanding of NAS. We have improved the writing to make clearer these prior debates and our intervention into them.
>
> The authors should more clearly present testable hypotheses, or if this is not possible, then suggest ways in which findings could be tested in future.
>
> * The core motivating hypothesis of this paper can be stated as: magnitude-based pruning in NAS will often select suboptimal architectures as the result of the interacting phenomena of operation bias and discretization error. Of course, it would be nice if we could present a simpler, punchier hypothesis of “magnitude-based pruning is often ineffective for architecture selection in NAS” however prior studies have already been published framed around this hypothesis. Instead, we modify our hypothesis to adopt a more subtle variant to highlight what our study can contribute that prior work cannot: a holistic explanation of why this is the case.  Again we hope that the new contributions section at the end of the Introduction helps to highlight these key takeaways and we appreciate your suggestion to better articulate them.

---

> > ### Comment · Reviewer_FmZd · 2023-05-08
> > **Improved abstract**
> >
> > I agree the authors improved the abstract and made contributions clearer.
> > My grade is 5, (previously 4.)